# Improving Decision Trees through the Lens of Parameterized Local Search

**Juha Harviainen**[*]
Department of Computer Science
University of Helsinki
Helsinki, Finland
`juha.harviainen@helsinki.fi`

**Frank Sommer**[*]
Institute of Computer Science
Friedrich-Schiller Universität Jena
Jena, Germany
`frank.sommer@uni-jena.de`

**Manuel Sorge**[*]
Institute of Logic and Computation
TU Wien
Vienna, Austria
`manuel.sorge@ac.tuwien.ac.at`

## Abstract

Algorithms for learning decision trees often include heuristic local-search operations such as (1) adjusting the threshold of a cut or (2) also exchanging the feature of that cut. We study minimizing the number of classification errors by performing a fixed number of a single type of these operations. Although we discover that the corresponding problems are NP-complete in general, we provide a comprehensive parameterized-complexity analysis with the aim of determining those properties of the problems that explain the hardness and those that make the problems tractable. For instance, we show that the problems remain hard for a small number $d$ of features or small domain size $D$ but the combination of both yields fixed-parameter tractability. That is, the problems are solvable in $(D+1)^{2d} \cdot |\mathcal{I}|^{\mathcal{O}(1)}$ time, where $|\mathcal{I}|$ is the size of the input. We also provide a proof-of-concept implementation of this algorithm and report on empirical results.

## 1 Introduction

Decision trees classify and explain data by associating labels with subsets of the feature space, characterized by a set of inequalities [30, 31, 35]. Not only are the models simple and efficient but also relevant for explainable AI because of their interpretability [15, 39]. Many prominent algorithms for learning decision trees like CART [2], C4.5 [36], C5.0, and J48 [43] utilize heuristics to decide when and where in the tree to add new internal nodes that *cut* the data based on some chosen feature and a threshold value. To mitigate overfitting, one may afterwards remove cuts from the decision tree by performing different types of pruning operations such as replacing some subtree by a leaf node [2, 36, 43]. Typically, when and where these are applied is also chosen heuristically.

After finding an initial solution a common and effective technique for optimization problems is to then perform *local search*, that is, searching for a better solution within some local neighborhood under some distance metric. Local search for decision trees is prevalent in the literature, in particular in simulated annealing approaches [6, 8], genetic algorithms as a mutator [20, 21, 22, 23, 27, 28, 29, 40], and SAT-based local search has recently shown strong performance for finding near-optimal decision trees [41]. However, local search is not applied in the standard decision-tree heuristics [2, 36, 43],

---

[*]Equal contribution.

39th Conference on Neural Information Processing Systems (NeurIPS 2025).

raising the question whether there is a justification for this situation. That is, what is the potential for local search to be effective for decision trees, in particular, heuristically computed ones? To answer this question, we need algorithms that are capable of, given a decision tree and training data, to find the *optimum* solution within a local neighborhood. Unfortunately, we are not aware of any systematic analysis of the algorithmics of this local-search problem. Contributing such an analysis is our main goal.

We focus on two natural types of local-search operations: Choose a cut and (1) *adjust* the threshold of the cut or (2) also *exchange* the feature associated with the cut. Both have been studied extensively in heuristics [6, 8, 20, 21, 22, 23, 27, 28, 29, 40]. We refer to searching the local neighborhood for an optimal modified tree as THRESHOLD ADJUSTMENT and CUT EXCHANGE. More specifically, we ask whether at most $k$ cuts can be modified (with adjustment or exchange) simultaneously such that the resulting tree has fewer errors than the input tree. As we show, both problems are NP-complete; thus we need more detailed algorithmic analysis to be relevant to practice. Thus, we identify the key parameters related to the local-search problems and characterize the influence of these parameters and most of their combinations on the complexity; see Figure 2. This continues a recent line of research on parameterized learning and improvement of decision trees [7, 10, 14, 24, 25, 33, 34, 42, 26].

There are three main levels of influence that a parameter $p$ can have when a problem is NP-hard: ideally (1) fixed-parameter tractability (FPT), that is, there is an algorithm with $f(p) \cdot |\mathcal{I}|^{\mathcal{O}(1)}$ running time, or (2) W[1]-hardness and XP-tractability, that is, there is an algorithm with running time $f(p) \cdot |\mathcal{I}|^{f(p)}$ and it is likely not possible to remove the dependence of the exponent on $p$, and (3) paraNP-hardness, that is, even for constant values of $p$ the problem is NP-hard. The parameters that we study are the number $n$ of training-data examples, number $d$ of features, the largest domain size $D$, the size $s$ of the input tree, the number $k$ of local search operations, the number $\ell$ of nodes unaffected by local search, an upper bound $t$ on the number errors that we aim for, and the largest number $\delta_{\max}$ of features in which two examples of different classes differ; see Figure 1. [2]

See Figure 2 for overviews over our results. Broadly, we observe that CUT EXCHANGE is at least as hard as THRESHOLD ADJUSTMENT; in particular this is justified by reductions from the latter THRESHOLD ADJUSTMENT to CUT EXCHANGE. Both are also related to the problem of learning an optimal decision tree from scratch: Some of our results can be interpreted to show hardness of learning decision trees even when we restrict the structure of the desired trees. Essentially, the associated reductions use scaffolding to construct a useless decision tree which then has to be made useful via changing its cuts with local-search operations where the number of local-search operations roughly corresponds to the size of the tree to be learned. (In practice, however, the number of local search operations will be lower than the tree size to be learned.) There are indeed parameter combinations where THRESHOLD ADJUSTMENT is strictly harder than learning decision trees, and CUT EXCHANGE is strictly harder than THRESHOLD ADJUSTMENT from a complexity point of view, see Figure 2.

On the positive side, several parameter combinations yield tractability: For instance while both problems are unlikely to be *fixed-parameter tractable* (FPT) in either the number $d$ of features or the maximum number $D$ of distinct values for a feature alone, their combination yields an FPT-algorithm. This algorithm can also straightforwardly incorporate the parameterized decision-tree pruning algorithm of Harviainen et al. [14] on the same parameters to improve the decision tree with local search and pruning at the same time, where the pruning operations either (1) *replace* a subtree by a new leaf or (2) *raise* a subtree $T_v$ rooted at $v$ by substituting a subtree rooted at an ancestor of $v$ with $T_v$ (see Theorem 4.3).

To complement our theoretical findings, we study the potential benefits of local search, that is, is there room for improvement in the decision trees constructed by the heuristics? While it is well known that optimally learning decision trees is NP-hard [16], the heuristics might still perform (almost) optimally on practical instances or at least be optimal in some local neighborhood of models. For instance, recent research [14] has—perhaps unexpectedly—suggested that the pruning heuristics employed by the commonly used libraries tend to be near-optimal on common benchmark datasets. We observe a similar phenomenon here suggesting near-optimality also locally, that is, while there is some room for improvement on the instances already by performing a single local search operation, the decrement in errors tends to be mild (see Table 1).

We proceed by fixing basic definitions and notation in Section 2. In Section 3, we show that learning decision trees reduces to local search. Section 4 focuses on algorithms for both problems, and in

---

[2]See Ordyniak and Szeider [34, Table 1] and Staus et al. [42, Table 3] for indication that $\delta_{\max}$ is small in relevant settings and see Harviainen et al. [14, Section 6] for parameters $d$ and $D$.

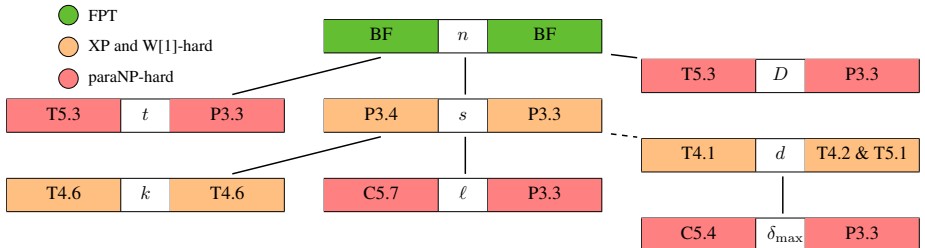

Figure 1: A Hasse diagram of the parameters, where an edge between two parameters means that the lower one is upper bounded by some function of the upper one. The dashed edge applies only to the THRESHOLD ADJUSTMENT problem. The color of the left box next to each parameter indicates the complexity for THRESHOLD ADJUSTMENT and the color of the right box for CUT EXCHANGE, which happen to be the same for all considered single parameters. The texts of the boxes refer to the corresponding theorems, and BF stands for a brute-force algorithm. The figure shows that for example, both problems are W[1]-hard and in XP when parameterized by $s$, and consequently they are at least W[1]-hard (potentially paraNP-hard) for all parameters smaller than $s$ (for example $\ell$) and have at least an XP-time algorithm (potentially FPT) for all parameters larger than $s$ (for example $n$).

Section 5 we provide further hardness results for both problems. Our main technical highlights are as follows: **(1)** For the FPT-algorithm for tree size $s$ and error bound $t$ for THRESHOLD ADJUSTMENT, in a first step we guess the changed cuts and the distribution of errors in the final tree and in a second step we rely on recursive binary-search to compute the optimal thresholds. Our other algorithms exploit structural properties via dynamic programming. **(2)** Our most involved hardness result is the W-hardness for the number $d$ (for both problems). We reduce from MULTICOLORED CLIQUE and create two features per color class and an input decision tree $T$ which has a regular structure. However, we need to be very careful with the values of the examples in these features (especially for THRESHOLD ADJUSTMENT since we cannot change the feature of any cut). In Section 6, we summarize our experiments and their results. Finally, we conclude with implications and potential research directions in Section 7. [3]

## 2   Preliminaries

In this section, we recall the basic definitions and terminology of decision trees and parameterized complexity. Denote the set $\{1, 2, \ldots, m\}$ by $[m]$ and the set $\{0, 1, \ldots, m\}$ by $[0, m]$.

**Decision trees.**   The set $E$ of $n$ labeled data points in the $d$-dimensional space $\mathbb{R}^d$ is called *examples*. The set of possible class labels of $E$ is denoted by $\Sigma$, which we assume to be $\{\mathsf{blue}, \mathsf{red}\}$ throughout the paper. The *value* of the $i$th *feature* of an example $e$ is $e[i]$ and the label of the example is $\lambda(e)$. The pair $(E, \lambda)$ is the *training data set*. For a feature $i$ and a *threshold* $x \in \mathbb{R}$, we define two subsets of examples $E_{\leq}[i, x] \coloneqq \{e \in E \colon e[i] \leq x\}$ and $E_{>}[i, x] \coloneqq E \setminus E_{\leq}[i, x] = \{e \in E \colon e[i] > x\}$.

A decision tree is a tuple $(T, \mathsf{feat}, \mathsf{thr}, \mathsf{cla})$ with a rooted tree $T$ on a set of nodes $V = V(T)$, functions $\mathsf{feat}$ and $\mathsf{thr}$ that associate a feature $\mathsf{feat}(v) \in [d]$ and a threshold $\mathsf{thr}(v) \in \mathbb{R}$ to each internal node $v \in V$, and a function $\mathsf{cla}$ that associates a class label $\mathsf{cla}(v) \in \Sigma$ for each leaf node $v \in V$. With minor abuse of terminology, we often call the rooted tree $T$ a decision tree without explicitly mentioning the functions $\mathsf{feat}$, $\mathsf{thr}$, and $\mathsf{cla}$. The internal nodes are also called *cuts*. Each cut $v \in V$ of the rooted tree $T$ has two ordered children which we refer to as its *left* and *right* children.

Each node $v \in V$ of the decision tree $T$ is then associated with a subset $E[T, v] \subseteq E$ of the examples, defined recursively as follows. For the root $r$ of $T$, we let $E[T, r] = E$. For a cut $v \in V$ whose left child is $u$ and right child is $w$, we let $E[T, u] = E[T, v] \cap E_{\leq}[\mathsf{feat}(u), \mathsf{thr}(u)]$ and $E[T, w] = E[T, v] \cap E_{>}[\mathsf{feat}(w), \mathsf{thr}(w)]$. For conciseness, let $E[v] \coloneqq E[T, v]$ when $T$ is clear from the context. Note that for any example $e$, the set of nodes $v$ for which $e \in E[v]$ is a path from the root of $T$ to one of its leaves $w$. We call $w$ the *leaf of* $e$ and say that $T$ assigns $e$ to the class $\mathsf{cla}(w)$. If $\lambda(e) = \mathsf{cla}(w)$, then $e$ is *correctly classified* (by $T$), and otherwise it is *misclassified* and called an *error*.

---

[3] A continuously updated version of our paper is available on arXiv [12] and the related source code for replicating the experiments on Zenodo [13].

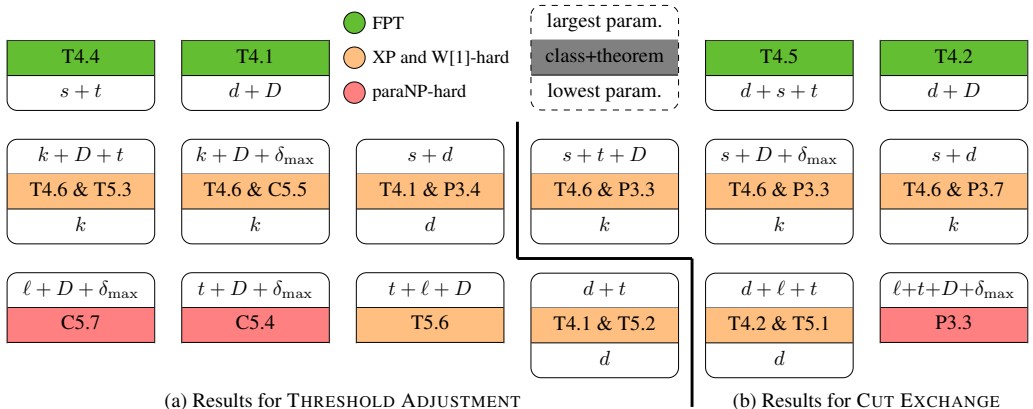

(a) Results for THRESHOLD ADJUSTMENT  (b) Results for CUT EXCHANGE

Figure 2: Summary of our results for combinations of parameters. In each box, the center cell indicates the complexity class (and references the corresponding theorem) which holds for all parameterizations bounded from below by the parameter of the bottom cell and from above by the parameter of the top cell. As a concrete example, the first box of the middle row states that THRESHOLD ADJUSTMENT is W[1]-hard and in XP for, for example, $k$, $k + D$, $k + t$, and $k + D + t$. Note that THRESHOLD ADJUSTMENT parameterized by $t + \ell + D$ is W[1]-hard but XP-tractability is open.

**Local search.** In the present work, we consider two natural local operations that modify the decision tree $(T, \mathsf{feat}, \mathsf{thr}, \mathsf{cla})$. The first operation is *threshold adjustment*, where we choose one cut $v \in V$ of the decision tree and a new threshold $x \in \mathbb{R}$, and obtain a new decision tree $(T', \mathsf{feat}', \mathsf{thr}', \mathsf{cla}')$ whose only difference to $(T, \mathsf{feat}, \mathsf{thr}, \mathsf{cla})$ is that $\mathsf{thr}'(v) = x$. Similarly in our other operation, *cut exchange*, we pick one cut $v \in V$, a new feature $i \in [d]$, and a new threshold $x \in \mathbb{R}$ to obtain a decision tree with $\mathsf{feat}'(v) = i$ and $\mathsf{thr}'(v) = x$. We also allow for arbitrarily changing the labels of the leaves of the decision tree to accommodate the modified cuts better.

Given these two local search operations, the question is then how much performing them some number of times helps us reduce the number of misclassifications, potentially after relabeling any number of leaves. We thus consider the following problems:

THRESHOLD ADJUSTMENT / CUT EXCHANGE

*Input:* A training data set $(E, \lambda)$, a decision tree $T$ for $(E, \lambda)$, and $k, t \in \mathbb{N}$.

*Task:* Perform $k$ threshold adjustments/cut exchanges to obtain a decision tree with at most $t$ errors.

**Reasonability.** To argue about the computational hardness of the problems in practice, we apply the notion of *reasonability* of Harviainen et al. [14]. Roughly speaking, a reasonable decision tree should be learnable by a heuristic algorithm for decision tree learning. More precisely, they propose two rules properties that a reasonable decision tree should satisfy: (1) no leaf $v$ should be empty, that is, $E[v] \neq \emptyset$, and (2) the class label of each leaf $v$ should match the majority of examples $e \in E[v]$.

**Parameterized complexity.** We study the complexities of these problems from the viewpoint of parameterized complexity [3, 4, 5], that is, we not only consider total encoding length $|\mathcal{I}|$ of the input instance $\mathcal{I}$ but also its properties. They are described by *parameters* $n$, $d$, $k$, and $t$ defined above, the *size* $s$ of the decision tree, the number of *unmodified cuts* $\ell := s - k$, the *domain size* $D := \max_{i \in [d]} |\{e[i] : e \in E\}|$ and $\delta_{\max}$, which is the maximum number of features where two examples of different classes have different values. These parameters have also been observed to be relevant in the literature for learning [10, 24, 34] and pruning [14] decision trees, and their relationships are illustrated in Figure 1. Note that $s \leq n$ by reasonability and that $d \leq n$ for threshold adjustment, since we can remove all features of the examples that do not appear in the initial decision tree.

A problem is *fixed-parameter tractable* (FPT) with respect to a parameter $p$ if any instance $\mathcal{I}$ of the problem is solvable in time $\mathcal{O}(f(p) \cdot \mathrm{poly}\,|\mathcal{I}|)$ for some function $f$. For example, FPT in $n$ for THRESHOLD ADJUSTMENT and CUT EXCHANGE follows by observing that there are at most $(s + 1)^n \leq n^n$ ways the examples can be partitioned to the leaves, and we can test in polynomial time for each partition whether it is obtainable with $k \leq s \leq n$ operations. A problem is *slicewise*

*polynomial* (XP) if it is solvable in time $\mathcal{O}(f(p) \cdot |\mathcal{I}|^{g(p)})$ for some functions $f, g$. Under the common assumption that FPT$\neq$W[$i$], $i \geq 1$, one can show that a problem is not in FPT for some parameter $p$ by reducing some known W[$i$]-hard problem to it in time $\mathcal{O}(f(p) \cdot \text{poly} \, |\mathcal{I}|)$.

Tighter lower bounds can be proven conditional to the exponential time hypothesis (ETH) [17, 18], which states that 3-SAT on $n$-variable formulas cannot be solved in $2^{o(n)}$ time. Essentially, one then shows that a fast algorithm to a problem would violate ETH through a reduction from 3-SAT to the problem. The strong exponential time hypothesis (SETH) [17, 18] similarly states that the satisfiability problem for $n$-variable formulas in conjunctive normal form cannot be solved in $\mathcal{O}((2 - \epsilon)^n)$ time.

## 3 From Computing Decision Trees to Local Search for Decision Trees

In this section we provide parameterized reductions from restricted versions of the computation of optimal decision trees to the associated local search problems for decision trees with respect to the two operations threshold adjustment and cut exchange, allowing us to transfer hardness results.

**Computation of minimal-size decision trees.** Initially, we reconsider the problem of computing a minimal-size decision tree DECISION TREE LEARNING and strengthen an existing hardness result which will also hold for one of our new restricted versions and subsequently also for CUT EXCHANGE. In DECISION TREE LEARNING the input consists of a training data set $(E, \lambda)$ and integers $s, t \in \mathbb{N}$, and the task is to compute a decision tree $T$ with at most $s$ inner nodes making at most $t$ errors. Gahlawat and Zehavi [10, Thm. 1] showed that DECISION TREE LEARNING is W[1]-hard for $s$ and cannot be solved in $|\mathcal{I}|^{o(s)}$ time unless the ETH fails, even if $\delta_{\max} = 3$. Here, we strengthen their result by ensuring that parameter $D$ is constant, while arguably being simpler.

**Theorem 3.1 (★).** DECISION TREE LEARNING *is W[1]-hard for $s$ and cannot be solved in* $|\mathcal{I}|^{o(s)}$ *time unless the ETH fails, even if $\delta_{\max} = 2$ and $D = 2$.*

**Computation of structure-restricted decision trees.** To obtain hardness results for CUT EXCHANGE, we reduce from the problem of computing a decision tree with a *fixed structure*, that is, the underlying tree of the decision tree $T$ is fixed. In other words, we are given a tree $Q$ and need to assign cuts to each inner node of $Q$. Moreover, for THRESHOLD ADJUSTMENT, we reduce from the problem of computing a decision tree with a *fixed structure* and a *fixed feature assignment*, that is, the function $\mathsf{feat}(\cdot)$ is given, mapping each inner node of the fixed tree $Q$ to a feature of $E$.

We consider the following two learning problems: (1) In FIXED STRUCTURE DECISION TREE (FS-DT) the input consists of a training data set $(E, \lambda)$, a tree $Q$, and $t \in \mathbb{N}$, and the task is to assign features and thresholds to all inner nodes of $Q$, and classes to the leaves such that the resulting decision tree $T$ makes at most $t$ errors. (2) In FIXED STRUCTURE FIXED FEATURE DECISION TREE (FSFF-DT) the input consists of a training data set $(E, \lambda)$, a tree $Q$, a function $\mathsf{feat}(\cdot)$ with fixed feature assignment, and $t \in \mathbb{N}$, and the task is to assign thresholds to all inner nodes of $Q$, and classes to the leaves such that the resulting decision tree $T$ makes at most $t$ errors.

Observe that FS-DT/FSFF-DT corresponds to the special case of CUT EXCHANGE/THRESHOLD ADJUSTMENT where $k = s$ if we assign some dummy cuts/thresholds to all inner nodes and some dummy classes to all leaves of the tree $Q$. We formalize that as the following observation and use it to obtain hardness results for both CUT EXCHANGE and THRESHOLD ADJUSTMENT via hardness results for FS-DT and FSFF-DT, respectively.

**Observation 3.2.** *There is a (parameterized) reduction from* FS-DT *to* CUT EXCHANGE *and from* FSFF-DT *to* THRESHOLD ADJUSTMENT *where $k = s$.*

**FS-DT and CUT EXCHANGE.** Note that in the reduction behind Theorem 3.1 the structure of any decision tree $T$ which is a solution is essentially fixed: The inner nodes of $T$ always form a path, exactly one leaf of the last cut in $T$ is blue, and all other leaves are red. Consequently, the reduction behind Theorem 3.1 also works for FS-DT. Since other hardness reductions for DECISION TREE LEARNING enforce similar structures of the resulting decision tree, we obtain the following.

**Proposition 3.3 (★).** *Even if the subgraph induced by the inner nodes is a path,* CUT EXCHANGE

1. *is W[1]-hard for $s$ and cannot be solved in $|\mathcal{I}|^{o(s)}$ time unless the ETH fails, even if $\ell = 0$, $D = 2$, and $\delta_{\max} = 3$.*

*2. is NP-hard even if $D = 2$, $\delta_{\max} = 2$, $\ell = 0$, and $t = 0$,*

*3. is W[2]-hard for $s$, cannot be solved in $|\mathcal{I}|^{o(s)}$ time unless the ETH fails, and cannot be solved in $\mathcal{O}(n^{s-\varepsilon})$ for any $\varepsilon > 0$ unless the SETH fails, even if $D = 2$, $\ell = 0$, and $t = 0$,*

**FSFF-DT and THRESHOLD ADJUSTMENT.** Next, for THRESHOLD ADJUSTMENT our goal is to show the following. (Running time lower bounds herein hold when assuming the ETH.)

**Proposition 3.4 ($\bigstar$).** *Even if the subgraph induced by the inner nodes is a path, $\ell = 0$, and $\delta_{\max}$ is a constant, THRESHOLD ADJUSTMENT is W[1]-hard wrt. $s + d$ and cannot be solved in $|\mathcal{I}|^{o(s+d)}$ time.*

To prove Proposition 3.4, we first take a closer inspection of a similar hardness result presented by Harviainen et al. [14, Thm. 5.6] for decision tree pruning with the so-called raising operation and show that their reduction can also be used to show a similar hardness result for FSFF-DT. Second, we use Observation 3.2 to obtain the result for THRESHOLD ADJUSTMENT.

**Proposition 3.5 ($\bigstar$).** *Even if the subgraph induced by the inner nodes is a path, $\ell = 0$ and $\delta_{\max}$ is a constant, FSFF-DT is W[1]-hard wrt. $s + d$ and cannot be solved in $n^{o(s+d)}$ time.*

**From THRESHOLD ADJUSTMENT to CUT EXCHANGE.** Finally, we provide a reduction from THRESHOLD ADJUSTMENT to CUT EXCHANGE which allows us to transfer our hardness result for parameter $s + d$ to CUT EXCHANGE. We achieve this in two steps: (1) We modify the THRESHOLD ADJUSTMENT instance $\mathcal{I}$ such that each feature is used at most once by adding new identical *original* features. (2) We replace each leaf $v$ of $\mathcal{I}$ by a new gadget $G_v$ consisting of a long path of cuts in new features and we add new dummy examples $E_v$ for each leaf $v$ which can only be correctly classified with our budget of allowed cut exchange operations if they end up in gadget $G_v$ corresponding to $v$. Thus, cut exchange operations performed at original cuts can only change their thresholds.

**Theorem 3.6 ($\bigstar$).** THRESHOLD ADJUSTMENT *reduces to* CUT EXCHANGE.

**Proposition 3.7 ($\bigstar$).** CUT EXCHANGE *is W[1]-hard with respect to $s + d$.*

# 4 Algorithms for THRESHOLD ADJUSTMENT and CUT EXCHANGE

Here, we provide FPT-algorithms and XP-algorithms for both problems and various parameters.

**The parameter $d + D$.** Now, we show that both problems admit an FPT-algorithm for $d + D$ via dynamic-programming. Simultaneously, these algorithms are also XP-algorithms for $d$. Afterwards, we show that this dynamic-programming approach also works if we allow both local search operations and also pruning. More precisely, in DECISION TREE PRUNING AND LOCAL SEARCH (DT-PLS) the input is a decision tree $T$ and we have 4 individual budgets, one for the number of threshold adjustment operations, one the number of cut exchange operations, one for the number of subtree replacement operations, and one for the number of subtree raising operations. Recall that in a subtree replacement operation an inner node $v$ of $T$ whose 2 children are leafs is replaced by a new leaf and in subtree raising the subtree rooted at an ancestor of $v$ is substituted by the subtree rooted at $v$.

**Theorem 4.1 ($\bigstar$).** THRESHOLD ADJUSTMENT *can be solved in $\mathcal{O}((D+1)^{2d+1}nk^2 + ns)$ time.*

*Proof Sketch.* We adapt the algorithm of Harviainen et al. [14] for subtree raising under the same parameterization. A *threshold sequence* $(l_i, r_i)_{i \in [d]}$ encodes for each feature $i \in [d]$ an interval $(l_i, r_i]$, characterizing a subset of the feature space. Let $E[(l_i, r_i)_{i \in [d]}] := \{e \in E : e[i] \in (l_i, r_i] \text{ for } i \in [d]\}$ be the set of examples within the box defined by a threshold sequence. Let $Q[v, (l_i, r_i)_{i \in [d]}, k']$ be the minimum number of errors achievable if our decision tree is the subtree rooted at $v \in V$ on our initial decision tree $T$ on the set of examples $E[(l_i, r_i)_{i \in [d]}]$ if we make at most $k'$ threshold adjustment operations. Define $L((l_i, r_i)_{i \in [d]}, j, x)$ as the threshold sequence identical to $(l_i, r_i)_{i \in [d]}$ except that for $i = j$ we have the interval $(\max\{l_i, x\}, r_i]$. Define $R((l_i, r_i)_{i \in [d]}, j, x)$ analogously with an interval $(l_i, \max\{r_i, x\}]$. For any cut $v$, when combining subresults of the left child $w$ and the right child $u$ of $v$, we either adjust $\mathsf{thr}(v)$ or not. Thus, $Q[v, (l_i, r_i)_{i \in [d]}, k']$ is the minimum of

$$\min_{\beta \in [0, k']} Q[w, L((l_i, r_i)_{i \in [d]}, \mathsf{feat}(v), \mathsf{thr}(v)), \beta] + Q[u, R((l_i, r_i)_{i \in [d]}, \mathsf{feat}(v), \mathsf{thr}(v)), k' - \beta], \text{ and}$$

$$\min_{\beta \in [0, k'-1], x \in \mathbb{R}} Q[w, L((l_i, r_i)_{i \in [d]}, \mathsf{feat}(v), x), \beta] + Q[u, R((l_i, r_i)_{i \in [d]}, \mathsf{feat}(v), x), k' - \beta - 1]. \quad \square$$

**Theorem 4.2 (★).** CUT EXCHANGE *can be solved in* $\mathcal{O}((D+1)^{2d+1}ndk^2 + ns)$ *time.*

**Theorem 4.3 (★).** DT-PLS *can be solved in* $\mathcal{O}((D+1)^{2d+1}ndk^8 + ns)$ *time.*

**The parameter $s + t$.** We now provide an FPT-algorithm for THRESHOLD ADJUSTMENT with respect to $s + t$ by iteratively using binary search to determine the new thresholds. I contrast, in Proposition 3.3 we showed that CUT EXCHANGE remains W[2]-hard with respect to $s$, even if $t = 0$. Hence, an FPT-algorithm for $s + t$ is unlikely. However, if we additionally include the number $d$ of features in the parameter, then we can obtain an FPT-algorithm.

**Theorem 4.4.** THRESHOLD ADJUSTMENT *can be solved in* $s^{\mathcal{O}(3s+t)} \cdot \text{poly}(|\mathcal{I}|)$ *time.*

*Proof.* Let $((E, \lambda), T, k, t)$ be an instance of THRESHOLD ADJUSTMENT and let $T'$ be a solution. Next, we describe our branch-and-bound algorithm: First, we guess the $k$ nodes $N$ where a threshold adjustment is done, that is, the $k$ nodes with different threshold in $T'$ than in $T$. Clearly, by testing all $\mathcal{O}(s^k) \in \mathcal{O}(s^s)$ node subsets of size at most $k$, in one branch the algorithm finds the correct node subset $N$ where the adjustments are performed. Moreover, since $T$ has $s + 1$ leaves, there are $2^{s+1}$ possible assignments of classes to the leaves. Again, by testing all $2^{s+1}$ of them, the correct one is found in one branch. In the following, we assume that the set $N$ and this assignment are fixed.

Second, we guess how many errors are done in each leaf in $T'$. Since $T$ has size $s$ and is binary, $T$ has at most $s + 1$ leaves. Now, for each of these leaves $i$ we guess a number $t_i \in [0, t]$ such that the sum of all these numbers is at most $t$. Let $\mathbf{t}$ be the corresponding *error vector*. Number $t_i$ represents the number of errors in leaf $i$ of the solution $T'$. Clearly, by testing all $\mathcal{O}(s^t)$ possibilities, in one branch the algorithm find the correct error distribution as in the solution $T'$. In the following, we assume that the error vector $\mathbf{t}$ is fixed.

Finally, we use binary-search to obtain the concrete thresholds of the nodes $N$. This technique is inspired by a similar procedure to obtain the thresholds when computing a minimal size (or minimal depth) decision tree [7, 10, 24, 34]. Consider a node $p \in N$ for which we have to find the correct threshold $t_p$ in $T'$. With the binary search, we determine the largest threshold value $t_p$ in the feature of node $p$ for which we can find thresholds for all nodes with unknown thresholds in the left subtree of $p$ respecting our guess on the number of errors in all leaves. Moreover, if for threshold $t_p$ we cannot find thresholds for all nodes with unknown thresholds in the right subtree of $p$ respecting our guess on the number of errors in all leaves, then we deal with a no instance: We cannot move threshold $t_p$ further to the left since then even more examples go to the right subtree of $p$ and thus the number of errors in the right subtree cannot decrease. The pseudocode of this binary search is shown in Algorithm 1. We defer the analysis of correctness and time complexity to the Appendix B.4. □

**Theorem 4.5 (★).** CUT EXCHANGE *can be solved in* $s^{\mathcal{O}(3s+t)}d^k \cdot |\mathcal{I}|^{\mathcal{O}(1)}$ *time.*

**The parameter $k$.** Finally, for the parameter $k$ we obtain XP-algorithms via brute-force.

**Proposition 4.6 (★).** *1.* THRESHOLD ADJUSTMENT *can be solved in time* $\mathcal{O}\left(s^{k+1}D^k n\right)$.

*2.* THRESHOLD ADJUSTMENT *can be solved in time* $\mathcal{O}\left((D+1)^s sn\right)$.

*3.* CUT EXCHANGE *can be solved in* $\mathcal{O}(((D+1)ds)^k ns)$ *time.*

## 5 Complementing Hardness Results for both Problems

First, we show that both problems are W[1]-hard for $d$ even if $t = 0$. This result implies that the parameter $s$ in Theorem 4.5 for CUT EXCHANGE cannot be dropped. Second, we take a closer inspection of THRESHOLD ADJUSTMENT. More specifically, we provide hardness results for various parameters including $k$. One of our results implies that the simple brute-force algorithm behind Proposition 4.6 for THRESHOLD ADJUSTMENT cannot substantially improved, unless the ETH fails.

**The parameter $d + t$.** We show that both problems are unlikely to admit an FPT-algorithm for $d+t$. Our results are inspired by a hardness reduction of Harviainen et al. [14, Thm. 5.10] who showed W[1]-hardness for decision tree pruning with subtree raising for the same parameter. Our reductions, however, especially the one for THRESHOLD ADJUSTMENT, are substantially more involved.

**Algorithm 1** Binary search to populate the unknown thresholds of the decision tree $T$. The initial call is `ComputeThresholds`$(E, r)$, where $E$ is the set of examples and $r$ is the root of $T$.

**Global Variables:** *Decision tree $T$, $N \subseteq V(T)$ of nodes with unknown threshold, error vector* **t**

**Function** `ComputeThresholds` *($E'$, $n$)*

    **Input:** An example set $E' \subseteq E$ and a node $n \in T$.

    **Output:** `True` if and only if we can assign thresholds to all nodes with unknown threshold in the decision tree corresponding to the subtree rooted at $n$ while respecting the error vector **t** if we only consider the examples $E'$.

| | | |
|---|---|---|
| 1 | **if** *n is a leaf* **then** | // check if error bound according to **t** is met |
| 2 |     Let $c$ be the class label of $n$ and $t_n$ be the entry of **t** corresponding to $n$ | |
| 3 |     **if** *more than $t_n$ examples of $E'$ do not have label $c$* **then return** `False` **else return** `True` | |
| 4 | **if** $n \in N$ **then** $z = $ `BinarySearch`$(E', n)$ | // $n$ has an unknown threshold |
| 5 | **else** $z = \mathsf{thr}(n)$ | |
| 6 | $p = $ right child of $n$,    $x = $ `ComputeThresholds`$(E'[f_{\mathsf{feat}(n)} > z], p)$ | |
| 7 | $q = $ left child of $n$,    $y = $ `ComputeThresholds`$(E'[f_{\mathsf{feat}(n)} \leq z], q)$ | |
| 8 | **return** $x$ AND $y$ | |

**Function** `BinarySearch` *($E'$, $n$)*

    **Input:** An example set $E' \subseteq E$ and a node $n \in T$.

    **Output:** The largest threshold $x$ of $\mathsf{feat}(n)$ s.t. we can populate all unknown thresholds in the left subtree of $n$ while respecting the error vector **t** if we only consider the examples $E'$.

| | |
|---|---|
| 9 | Set $D$ to be an array containing all thresholds of feature $\mathsf{feat}(n)$ in ascending order |
| 10 | Set $L = 0$, $R = |\mathtt{length}(D)| - 1$, $b = 0$,    $q = $ left child of $n$ |
| 11 | **while** $L \leq R$ **do** |
| 12 |     $m = \lfloor (L+R)/2 \rfloor$ |
| 13 |     **if** `ComputeThresholds`$(E'[f_{\mathsf{feat}(n)} \leq D[m]], q) = $ `True` **then** $L = m+1$, $b = 1$ |
| 14 |     **else** $R = m - 1$, $b = 0$ |
| 15 | **if** $b = 1$ **then return** $D[m]$ |
| 16 | **return** $D[m-1]$, with $D[-1] = D[0] - 1$ |

For both problems, we reduce from MULTICOLORED INDEPENDENT SET, where the input is a graph $G$, and $\kappa \in \mathbb{N}$, where the vertex set $V(G)$ of $N$ vertices is partitioned into $V_1, \ldots, V_\kappa$. The question is whether $G$ contains an independent set consisting of exactly one vertex per class $V_i$. MULTICOLORED INDEPENDENT SET is W[1]-hard parameterized by $\kappa$ and cannot be solved in $f(\kappa) \cdot n^{o(\kappa)}$ time unless the ETH fails [3]. In both reductions, we create two features $d_i^<$ and $d_i^>$ per class $i$ such that all cuts in features $d_i^<$ and $d_i^>$ together correspond to a vertex selection in $V_i$. We achieve this as follows: For each pair $d_i^<$ and $d_i^>$, we create examples which can only be separated in these two features and which have labels blue and red alternatingly. Hence, for each possible threshold $x$ in features $d_i^<$ and $d_i^>$, the resulting decision tree $T$ needs to contain either cut $(d_i^<, x)$ or cut $(d_i^>, x)$. Furthermore, for each edge we create a red *edge example*. If vertex $v_{i,j} \in V_i$ is selected, then all edge examples corresponding to edges having an endpoint in color class $i$ which is *not* $v_{i,j}$ will then be correctly classified by the resulting decision tree $T$. Thus, we can only correctly classify an edge example if we *do not* select at least one endpoint of the corresponding edge.

For CUT EXCHANGE the input decision tree $T'$ is a path with red and blue leaves alternatingly with cuts in a dummy feature $\widehat{d}$. To obtain $T$, all cuts of $T'$ which are in $\widehat{d}$, need to be changed to cuts in features $d_i^<$, $d_i^>$, and $d_i^*$. For THRESHOLD ADJUSTMENT it is more complicated since we cannot change the feature of any cut: Thus, we extend features $d_i^<$ and $d_i^>$ by new dummy thresholds which are used in the input decision tree $T'$. Since $t = 0$, however, $T'$ cannot be a path anymore. Instead, we add new cuts in dummy cuts in new dummy features which cannot be adjusted, as we show.

**Theorem 5.1 (★).** CUT EXCHANGE *is W[1]-hard for $d$ even if $\delta_{\max} = 6$, $\ell = 0$ and $t = 0$. Under ETH, the problem cannot be solved in time $|\mathcal{I}|^{o(d)}$.*

**Theorem 5.2 (★).** THRESHOLD ADJUSTMENT *is W[1]-hard for $d$ even if $t = 0$ and $\delta_{\max} = 20$. Under ETH, the problem cannot be solved in time $|\mathcal{I}|^{o(d)}$.*

**Other hardness results for THRESHOLD ADJUSTMENT.** Now, we prove hardness for many parameters including $k$. Afterwards, we present two hardness results for parameters including $\ell + D$.

**Theorem 5.3 (★).** THRESHOLD ADJUSTMENT *is W[2]-hard for $k$ even if $D = 2$ and $t = 0$. Under the ETH, the problem cannot be solved in time $\mathcal{O}(|\mathcal{I}|^{o(k)})$, and under the SETH, the problem cannot be solved in time $\mathcal{O}(s^{k-\epsilon})$ for any $\epsilon > 0$.*

*Proof Sketch.* We reduce from the W[2]-hard problem HITTING SET [3]. Consider an instance of HITTING SET with a universe $U = \{1, 2, \ldots, |U|\}$ and a family of sets $\mathcal{S}$ such that we look for a hitting set of size $\kappa$. We create a THRESHOLD ADJUSTMENT instance with $|U|$ features $d_1, d_2, \ldots, d_{|U|}$, and for each set $S$ in $\mathcal{S}$ we create a red example $e$ such that $e[d_i] = 0$ if $i \in S$, and $e[d_i] = 1$ otherwise. We also create one blue example $e'$ with $e'[d_i] = 1$ for all $i \in U$. Our initial decision tree has $|U|$ cuts $c_1, c_2, \ldots, c_{|U|}$ that form a chain where the right child of the cut $c_i$ is $c_{i+1}$ for all $i \in [|U| - 1]$. The right child of $c_{|U|}$ is a blue leaf. The left child of each cut is a red leaf. The feature of the cut $c_i$ is $d_i$ and the initial threshold is 0. Finally, let $k = \kappa$ and $t = 0$. The correctness proof, the lower bound, and adaptation for reasonable trees are deferred to the appendix. □

Our next two results follow by reducing from (PARTIAL) VERTEX COVER instead of HITTING SET.

**Corollary 5.4 (★).** THRESHOLD ADJUSTMENT *is NP-hard even if $D = 3$, $\delta_{\max} = 2$, and $t = 0$.*

**Corollary 5.5 (★).** THRESHOLD ADJUSTMENT *is W[1]-hard for $k$ even if $D = 3$ and $\delta_{\max} = 2$.*

To achieve our final two results, we again reduce from HITTING SET/VERTEX COVER. Each cut starts in an *undecided* state (threshold 0) and has to be *included* (threshold 1) or *not included* (threshold −1) in the hitting set. Including a cut yields 1 error and thus only a fixed number of cuts can be activated.

**Theorem 5.6 (★).** THRESHOLD ADJUSTMENT *is W[2]-hard for $t$ even if $\ell = 0$, $D = 3$. Under ETH, the problem cannot be solved in time $\mathcal{O}(|\mathcal{I}|^{o(t)})$, and under SETH, the problem cannot be solved in time $\mathcal{O}(s^{t-\epsilon})$ for any $\epsilon > 0$.*

**Corollary 5.7 (★).** THRESHOLD ADJUSTMENT *is NP-hard even if $\ell = 0$, $D = 3$, $\delta_{\max} = 3$.*

## 6 Experiments

The goals of our small-scale empirical study were as follows: (1) Assess whether our algorithms can in principle be used to study the local-search neighborhoods of decision trees that are computed from real-world benchmark data by established heuristics. (2) Evaluate the potential performance of local search in optimizing existing heuristic decision trees. (3) Assess whether performing pruning and local search simultaneously is superior to the heuristics in terms of classification error. The experiments are not designed to be conclusive; instead they have a proof-of-concept nature and aim to spur follow-up work.

We used 47 datasets from the Penn Machine Learning Benchmarks library [38], including 32 previously used for minimum-size tree computation [1, 32, 42] and additional larger datasets. The datasets range from 72 to 5404 examples (mean 655.09, median 302.00), see the appendix for more details. We computed unpruned and pruned trees using WEKA 3.8.5's [9] C4.5 implementation [36] J48. For the pruned trees we used J48's subtree replacement heuristic; more details in the appendix.

We implemented an algorithm based on Theorem 4.3 that, given a dataset and a decision tree $T$, can compute for every tuple $(k_{\mathsf{ad}}, k_{\mathsf{ex}}, k_{\mathsf{re}})$ the minimum number of errors attainable by a tree obtained from $T$ with at most $k_{\mathsf{ad}}$ adjustment, $k_{\mathsf{ex}}$ exchange, and $k_{\mathsf{re}}$ subtree replacement operations (where $k_{\mathsf{re}}$ counts the number of cut nodes). Since naive enumeration of all threshold sequences in the dynamic program is wasteful, we instead start, for each node $v$ in the tree, with the sequence of thresholds implied by the cuts above $v$ and then compute all modifications of this sequence obtainable by the remaining number of adjustment and replacement operations above $v$. This maintains optimality. Nevertheless, there is large potential for heuristic improvements. We implemented the algorithm in Python 3.10.12 and all experiments were carried out on a personal computer running Ubuntu Linux 22.04 with an 8-core Intel Core i7-9700 processor and 16GiB RAM (running up to eight instances of the algorithm in parallel).

We first checked whether the heuristically computed real-world pruned decision trees are locally optimal. That is, for each of the 47 datasets we took the pruned tree and checked whether their misclassifications could be reduced by up to two adjustment or exchange operations. For adjustment, 33 (70.21%) thus obtained problems could be solved within a 1h time limit, for exchange it was 17 (36.17%). The heuristically computed trees were 2-optimal or close to 2-optimal, that is, they

Table 1: Error rates for different operations. Dashes indicate timeouts.

| Dataset | Initial | $k_{\mathsf{ad}} = 1$ | $k_{\mathsf{ad}} = 2$ | $k_{\mathsf{ex}} = 1$ | $k_{\mathsf{ex}} = 2$ |
|---|---|---|---|---|---|
| banana | 249 | 248 | – | – | – |
| breast-cancer | 31 | 30 | 30 | 30 | 29 |
| cleve | 15 | 14 | 14 | 14 | – |
| cleveland | 13 | 12 | 12 | 12 | – |
| cleveland-nominal | 23 | 23 | 23 | 22 | 22 |
| colic | 15 | 15 | 15 | 14 | – |
| ecoli | 10 | 8 | 8 | 8 | 8 |
| heart-c | 15 | 14 | 14 | 14 | – |
| lymphography | 5 | 5 | 5 | 5 | 4 |
| profb | 42 | 41 | – | 41 | – |

can hardly be improved with up to two local-search operations: with at most two exchanges, 9 trees could be improved, and with up to two adjustments, 7 trees. The reduction in misclassifications was at most 2. Table 1 shows the trees which could be improved. To summarize, for small number of local search operations the implementation is feasible and it seems that heuristically computed trees are close to being locally optimal.

We also looked at combining pruning with local search. We first investigated whether local search enables further pruning heuristically pruned trees: For 18 datasets, this was possible after adjustments (avg. 1.40 nodes) and for 24 after exchanges (avg. 1.78 nodes)—with one tree size nearly halving after a single exchange. Starting with unpruned trees and combining pruning with local search gave better results than sequential application: 10 and 17 trees showed error reductions of 5.16% and 8.92% on average, respectively. We also computed full pareto fronts for the tradeoff between pruning and errors (Figure 7, appendix); here local search becomes more effective with aggressive pruning. This indicates that heuristics typically select tradeoffs minimally amenable to local search; full details in the appendix. Overall, our algorithms provide new insights indicating local search becomes increasingly valuable with more extensive pruning than commonly applied. Note that to obtain the above results we need algorithms that find the optimum, which we here designed for the first time.

## 7 Outlook

We gave a comprehensive analysis of the parameterized complexity of improving decision trees with the local search operations threshold adjustment and cut exchange. We presented both algorithmic results and complexity-theoretic lower bounds for both problems corresponding to the two operations. Based on these algorithms we provided a proof-of-concept implementation and demonstrated that there is some potential—albeit limited—for improvement in the trees constructed by heuristics. The experiments were hampered by the computational hardness of the problems, which might be mitigable by designing new approaches to them, for example, by formulating them as a mixed integer program.

While we settled the complexity for many parameter combinations for both problems, some combinations are still open. For example is THRESHOLD ADJUSTMENT FPT with respect to $d + \ell + t$, or is CUT EXCHANGE FPT with respect to $d + k + t$? Investigating so-called ensembles of decision trees under the local search operations threshold adjustment and cut exchange is an interesting direction for future research. Clearly, our hardness result transfer to ensembles, but for the algorithms this is not clear. Moreover, on the practical side it is interesting to investigate whether the effect of local search for ensembles is stronger than for a single decision tree. Also, it is interesting to study the complexity of other local search operations for decision trees which change the structure of the tree, for example subtree switching [37] or subtree substitution [41]. Moreover, since our new approach yields optimal solutions for local search of decision trees, our algorithms could be used to develop new genetic algorithms with more powerful crossover operations. Finally, currently minimal-size decision trees can only be computed for size up to $s = 20$ [42] and a major hurdle are good lower bounds to abort parts of the branch-and-bound approach early. Our local search algorithm could help in strengthening existing lower bounds and thus make the learning of minimal-size decision trees more applicable.

## Acknowledgments and Disclosure of Funding

Juha Harviainen was supported by the Research Council of Finland, Grant 351156. We also gratefully acknowledge support by the TU Wien International Office for hosting Juha Harvianen in Vienna.

Frank Sommer was supported by the Alexander von Humboldt Foundation and partially by the Carl Zeiss Foundation, Germany, within the project "Interactive Inference".

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

# Appendix

## A   Additional Material for Section 3

### A.1   Proof of Theorem 3.1

*Proof.* We reduce from PARTIAL VERTEX COVER where we are given a graph $G$ and two integers $\kappa$ and $\tau$ and the goal is to select at most $k$ vertices $S$ of $G$ such that at least $\tau$ edges are covered by $S$. PARTIAL VERTEX COVER is W[1]-hard with respect to $\kappa$ [11], and unless the ETH fails, PARTIAL VERTEX COVER cannot be solved in $f(\kappa) \cdot n^{o(\kappa)}$ time [3].

We create one binary feature $d_v$ for each vertex $v \in V(G)$. For each edge $e = uw \in E(G)$ we create one red example $e_{uw}$ which has value 1 in the two features $d_u$ and $d_w$ which are corresponding to its endpoints $u$ and $w$, and value 0 in each other feature. Moreover, we add blue examples $b_1, \ldots, b_{n^2}$ which has value 0 in each feature. Finally, we set $s := \kappa$ and $t := m - \tau$, where $m := |E(G)|$. Observe that $\delta_{\max} = 2 = D$. Note that all blue examples have the same coordinates in each feature. In the end we argue how we can modify the instance such that each pair of (blue) examples differs in at least one feature.

We now show that $G$ has a vertex set $S$ of size at most $\kappa$ covering $\tau$ edges if and only if there exists a decision tree $t$ of the DECISION TREE LEARNING instance of size $s$ with at most $t$ errors.

($\Rightarrow$) Let $S$ be a solution of $(G, k, \tau)$. Let $T$ be the decision tree whose inner nodes are a path and where there is a one-to-one correspondence between vertices in $S$ and the cuts of the form $d_x > 0$ which are in $T$. Each left subtree of these cuts leads to a red leaf and the unique right leaf is blue. Now, observe that each red edge example $e_{uw}$ corresponding to an edge $uw$ which has at least one endpoint in $S$ is put in one of the red leaves. Moreover, all blue examples end up in the unique blue leaf. Since at least $\tau$ edges are incident with at least one vertex of $S$, we conclude that $T$ has the desired properties.

($\Leftarrow$) Let $T$ be a minimal decision tree with at most $s = \kappa$ inner nodes making at most $t = m - \tau$ errors. Since all blue examples have the same coordinates in all features, it is safe to assume that the inner nodes of $T$ are a path such that exactly one leaf of the last cut in $T$ is a blue leaf and all other leaves are red. Moreover, since each feature is binary, we conclude that all cuts of $T$ use different features. Now let $S$ be the set of vertices corresponding to features used in $T$. Since $T$ makes at most $t = m - \tau$ errors, and since all $n^2 > m$ blue examples are indistinguishable, at most $t = m - \tau$ red examples are misclassified by $T$. Consequently, $\tau$ red edge examples are correctly classified by $T$. By construction each edge corresponding to one of these edge examples has at least one endpoint in $S$ and thus $S$ covers at least $\tau$ edges of $G$. $\qquad\square$

### A.2   Proof of Proposition 3.3

*Proof.* First, observe that all three results hold for FS-DT: 1. holds due to Theorem 3.1. Similarly, hardness results shown for computing minimal-size decision trees were the induced subgraph of the inner nodes form a path and exactly one leaf is blue and all other leafs are red (or vice versa) by reductions from VERTEX COVER and HITTING SET instead of PARTIAL VERTEX COVER are known. More precisely, 2. was shown by Ordyniak and Szeider [34, Thm. 2], and 3. was proven by Ordyniak and Szeider [34, Thm. 3]. Second, together with Observation 3.2 we obtain analog hardness results for CUT EXCHANGE. $\qquad\square$

### A.3   Proof of Proposition 3.5

*Proof.* By a reduction from MULTICOLORED CLIQUE, Harviainen et al. [14, Thm. 5.6] showed that DTRAIS$_=$ is W[1]-hard for $d + \ell$ even if $\delta_{\max} = 6$. In DTRAIS$_=$ the input consist of a classification instance, a decision tree $T$ with $s'$ inner nodes and two integers $t$ and $\ell$. The goal is to prune exactly $s' - \ell$ inner nodes of $T$ with the so-called raising operation such that the resulting decision tree $T'$ consisting of $\ell$ inner nodes makes at most $t$ errors. Moreover, in MULTICOLORED CLIQUE the input consists of a $\kappa$-partite graph $G$ where each partite set consists of $q$ vertices, and the goal is to find a clique of size $\kappa$ having exactly one vertex in each partite set. In their construction they create 2 features per partite set of $G$, that is, $2 \cdot \kappa$ features in total. Moreover, the input decision tree $T$ has a very restrictive structure: (1) The subgraph induced by the inner nodes is a path $(p_1, p_2, \ldots, p_z)$,

where $z = q \cdot 2 \cdot \kappa$. (2) $T$ has a unique red leaf; in other words, all leafs except one are blue. (3) Inner nodes $p_{1+i\cdot q}, \ldots, p_{q+i\cdot q}$ correspond to possible values of feature $i$ for each $i \in [0, 2 \cdot \kappa - 1]$. Finally, they set $\ell = 2 \cdot \kappa$. In the correctness part of their proof, Harviainen et al. [14, Thm. 5.6] argue that exactly one cut in each feature has to remain to fulfill to error bound $t$.

Hence, the reduction of Harviainen et al. [14, Thm. 5.6] can easily be adapted for FSFF-DT by giving as input a decision tree $T$ with exactly $\ell = 2 \cdot \kappa$ inner nodes which are arranged in a path and each of these inner nodes is equipped with a unique feature of the $2 \cdot \kappa$ total features. The correctness can be shown similar as for DTRAIS$_=$; for FSFF-DT it is simpler since we do not need to argue that exactly one cut per feature remains. Also, the leaf-assignment cannot be changed due to the so-called forcing examples. □

### A.4 Proof of Proposition 3.4

*Proof of Proposition 3.4.* The proof is a immediate consequence of Proposition 3.5 and Observation 3.2. □

### A.5 Proof of Theorem 3.6

*Proof.* The main idea is to add sets of new examples for each leaf of the decision tree from THRESHOLD ADJUSTMENT such that at least $t + 1$ of those examples gets misclassified if the feature of some original cut is changed. We achieve this, by adding a gadget to each leaf of the THRESHOLD ADJUSTMENT instance. Moreover, our budget is chosen such that we can and have to perform exactly one cut exchange in each of these gadgets.

**Construction.** We first show that we can assume without loss of generality that no feature appears twice in the initial decision tree.

For all features $d_i$, replace the first occurrence of the feature $d_i$ by a new feature $d_i^1$, the second by $d_i^2$, and so on. Modify the examples $e \in E$ accordingly, that is, let $e[d_i^j] = e[d_i]$ for the new features $d_i^j$. Now, the original instance to THRESHOLD ADJUSTMENT has a solution if and only if our modified instance to THRESHOLD ADJUSTMENT has a solution.

We then start constructing our CUT EXCHANGE instance by using this THRESHOLD ADJUSTMENT instance as our starting point. For the budget $k$ of allowed cut exchange operations, we set $k := k' + s' + 1$, where $k'$ is the budget of allowed threshold adjustment operations and $s'$ is the size of the THRESHOLD ADJUSTMENT instance. Also, all examples of the initial THRESHOLD ADJUSTMENT instance are referred to as *original*.

Moreover, we index the leaves arbitrarily. For each leaf $v$ with index $j \in [s' + 1]$, we create $2(t+1)(k+1) + 2(n+t+2) + 1$ new *leaf examples* whose leaf is initially $v$ and remains $v$ if only threshold adjustments are performed. Recall that $n$ is the number of examples in the THRESHOLD ADJUSTMENT instance. Consider the root-to-leaf path $v_1, v_2, \ldots, v_m$ for $v$, where $v_m = v$. Then, add $k$ red leaf examples $e_{j,1}^+, e_{j,2}^+, \ldots, e_{j,k}^+$, $k$ red leaf examples $e_{j,1}^-, e_{j,2}^-, \ldots, e_{j,k}^-$, and two blue leaf examples $e_{j,0}^+, e_{j,0}^-$. For each of these leaf examples, we create $t + 1$ copies. Moreover, we create two red leaf examples $e_{j,k+1}^+, e_{j,k+1}^-$. Next, we create $n + t + 2$ copies of both of these examples. Finally, we add another blue leaf example $e_j$.

Now, we define the values in each feature. All features which are constructed so far are referred to as the *original features*. First, we set the coordinates of all leaf examples in the original features as follows: If $v_{i+1}$ is the left child of $v_i$, we set $e_j[\mathsf{feat}(v_i)] = -\infty$ and for all $l \in [0, k+1]$ we let $e_{j,l}^+[\mathsf{feat}(v_i)] = e_{j,l}^-[\mathsf{feat}(v_i)] = -\infty$, and otherwise, we set $e_j[\mathsf{feat}(v_i)] = \infty$ and we let $e_{j,l}^+[\mathsf{feat}(v_i)] = e_{j,l}^-[\mathsf{feat}(v_i)] = \infty$. Moreover, for each other original feature $d'$, that is, $d' \neq \mathsf{feat}(v_i)$ for all $v_i$ on the path to leaf $v$, we set $e_j[d'] = -\infty$ and for all $l \in [0, k+1]$ we let $e_{j,l}^-[d'] = -\infty$, and $e_{j,l}^+[d'] = \infty$.

We continue by creating $k + 2$ new *gadget features* $d_{j,1}, d_{j,2}, \ldots, d_{j,k}$, and $d_{j,\alpha}, d_{j,\beta}$ for each leaf $v$ with index $j$. We set $e_{j,l}^+[d_{j,l}] = e_{j,l}^-[d_{j,l}] = 0$ for $l \in [k]$. Moreover, we set $e_{j,0}^+[d_{j,\alpha}] = e_{j,0}^-[d_{j,\alpha}] = e_{j,0}^+[d_{j,\beta}] = e_{j,0}^-[d_{j,\beta}] = 0$ and we set $e_j[d_{j,\alpha}] = e_j[d_{j,\beta}] = 0$. For each other combination of a leaf example $e$ and a gadget feature $d'$, we set $e[d'] = \infty$. It remains to define the coordinates of

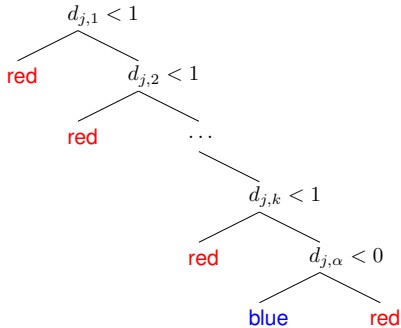

Figure 3: The gadget used for misclassifying leaf examples that do not correspond to the $j$th leaf $v$ in Theorem 3.6. Note that the class of $v$ is not relevant for this gadget.

the original examples in the gadget features. For each original example $e$ we do the following: We set $e[d_{j,\alpha}] = 0$ for each $j \in [s' + 1]$. For each remaining gadget feature $d'$ we set $e[d'] = \infty$.

As the last step of the reduction, we replace the $j$th leaf $v$ of the initial decision tree by a gadget that is a chain of $k + 1$ inner nodes as follows: The chain has $k + 1$ cuts in features $d_{j,1}, d_{j,2}, \ldots, d_{j,k}, d_{j,\alpha}$ with threshold value 1 for the cuts in features $d_{j,l}$ for $l \in [k]$ and red leaves as the left children, except for the cut in feature $d_{j,\alpha}$ which uses threshold 0 and that has a blue leaf as the left child and a red leaf as the right child. This is illustrated in Figure 3. Call the added cuts *gadget cuts* and the other cuts *original cuts*.

Observe that the so-constructed tree is reasonable: (1) The left child of cut $d_{j,l} < 1$, which is a red leaf contains the red leaf examples $e_{j,l}^+, e_{j,l}^-$. (2) The left child of cut $d_{j,\alpha} < 0$, which is a blue leaf contains the blue leaf examples $e_j$. (3) The right child of cut $d_{j,\alpha} < 0$, which is a red leaf contains the red leaf examples $e_{j,k+1}^+, e_{j,k+1}^-$, the blue leaf examples $e_{j,0}^+, e_{j,0}^-$, and the original examples ending up at the $j$th leaf. Since there are $n$ original examples, for examples $e_{j,0}^+, e_{j,0}^-$ we have $t + 1$ copies each, and for examples $e_{j,k+1}^+, e_{j,k+1}^-$ we have $n + t + 2$ copies each, we conclude that this leaf contains more red examples than blue examples. Hence, this tree is reasonable.

**Correctness.** We claim that this instance of CUT EXCHANGE has a solution with $k = k' + s' + 1$ cut exchanges if and only if the original THRESHOLD ADJUSTMENT instance has a solution with $k'$ threshold adjustments.

($\Rightarrow$) Suppose that a solution to the THRESHOLD ADJUSTMENT instance $I$ exists. In the first step, we change the corresponding original cuts in the CUT EXCHANGE instance by assigning the thresholds of the solution to $I$. Second, for each leaf $v$ of the solution to $I$, we do the following: If $v$ is a blue leaf, then we change the cut $d_{j,\alpha} < 0$ to $d_{j,\alpha} < 1$, and otherwise, if $v$ is a red leaf, we change the cut $d_{j,\alpha} < 0$ to $d_{j,\beta} < 1$. Clearly, we perform at most $k = k' + s' + 1$ cut exchange operations. Observe that after these cut exchanges all leaf examples are correctly classified: (1) The red leaf examples $e_{j,l}^+, e_{j,l}^-$ end up in the left child of cut $d_{j,l} < 1$ which is a red leaf. (2) The blue leaf examples $e_{j,0}^+, e_{j,0}^-, e_j$ end up in the left child of cut $d_{j,\alpha} < 1$ or $d_{j,\beta} < 1$, respectively, which is a blue leaf. (3) The red leaf examples $e_{j,k+1}^+, e_{j,k+1}^-$ end up in the right child of cut $d_{j,\alpha} < 1$ or $d_{j,\beta} < 1$, respectively, which is a red leaf.

Hence, it remains to consider the original examples: Consider the $j$th leaf in the solution to $I$. If $v$ is blue, then we replaced cut $d_{j,\alpha} < 0$ with cut $d_{j,\alpha} < 1$. Consequently, all examples ending in $v$ in the solution to $I$ end up in the left child of $d_{j,\alpha} < 1$ which is a blue leaf. Otherwise, if $v$ is red, then we exchanged cut $d_{j,\alpha} < 0$ with cut $d_{j,\beta} < 1$. Consequently, all examples ending in $v$ in the solution to $I$ end up in the right child of $d_{j,\beta} < 1$ which is a red leaf. Thus, the same set of original examples is correctly classified and thus we have at most $t$ errors in total.

($\Leftarrow$) Suppose now instead that a solution to the CUT EXCHANGE instance exists. First, note that to separate the red leaf examples $e_{j,1}^+, e_{j,2}^+, \ldots, e_{j,k}^+$ and $e_{j,1}^-, e_{j,2}^-, \ldots, e_{j,k}^-$ from the blue leaf examples $e_{j,0}^+$ and $e_{j,0}^-$, the tree needs to contains the cuts $d_{j,l} < 1$ for each $l \in [k]$ since feature $d_{j,l}$ is the unique feature in which $e_{j,l}^+, e_{j,l}^-$ differ from $e_{j,0}^+, e_{j,0}^-$. Since there are $t + 1$ copies of each leaf

examples, lack of such a cut causes at least $t + 1$ errors. Moreover, in order to separate the blue leaf examples $e_{j,0}^+, e_{j,0}^-$ from the red leaf examples $e_{j,k+1}^+, e_{j,k+1}^-$ the tree needs to contain either the cut $d_{j,\alpha} < 1$ or $d_{j,\beta} < 1$ since these examples only differ in features $d_{j,\alpha}, d_{j,\beta}$. Again, since for each of these examples we have at least $t + 1$ copies, lack of such a cut causes at least $t + 1$ errors. Moreover, observe that the cut $d_{j,\alpha} < 1$ or $d_{j,\beta} < 1$ has to appear after any cut in feature $d_{j,l}$ since otherwise the blue leaf examples $e_{j,0}^+, e_{j,0}^-$ cannot be separated from the red leaf examples $e_{j,l}^+, e_{j,l}^-$, causing at least $t + 1$ errors. Furthermore, note that by the fact in which all of these leaf examples end up, the leaf labeling of the gadget is fixed, that is, is identical to the initial labeling shown in Figure 3.

Note that since the initial tree neither contains the cut $d_{j,\alpha} < 1$ nor the cut $d_{j,\beta} < 1$, this consumes $s' + 1$ cut exchange operations of our budget of $k = k' + s' + 1$. Consequently, the remaining budget for the number of cut exchange operations is $k'$.

Assume now that we changed the feature of some inner node $z$ (except the case that $z$ has a cut $d_{j,\alpha} < 0$) to $d^*$ in our solution to CUT EXCHANGE, and without loss of generality, assume that the features on the inner nodes between the root and $z$ remain the same (but the thresholds may change). There are two cases: either (1) $z$ is an original cut or (2) $z$ is a gadget cut.

(1): Let $w$ be a child of $z$ whose subtree does not have the feature $d^*$ and let $j$ be the index of some leaf there. Such a child exists, since each cut has a unique feature in the initial decision tree. Let $u$ be the other child of $z$. Consider the case where $w$ is the left subtree of $z$; the proof is analogous for the right subtree. In the initial decision tree $T$, we have $e_{j,l}^+ \in E[T, w]$ for all $l \in [k + 1]$. However, $e_{j,l}^+[d^*] = \infty$, and thus $e_{j,l}^+ \in E[T', u]$ in the modified decision tree $T'$. Now, we have budget to perform at most $k - 1$ additional cut exchange operations, and therefore there is an example $e_{j,l'}^+$ that ends up at the same leaf as $e_{j,0}^+$, since none of the features $d_{j,1}, d_{j,2}, \ldots, d_{j,k}$ appear in the subtree rooted at $u$. This results in at least $t + 1$ misclassifications because there are $t + 1$ copies of both examples. Hence, no feature in an original cut can be changed without yielding at least $t + 1$ errors.

(2): Assume that $z$ has a gadget cut different from $d_{j,\alpha} < 0$. Note that the features on that path from root to $z$ were not changed, and thus $E[T, z]$ and $E[T', z]$ contain the same set of leaf examples for the original decision tree $T$ and the modified decision tree $T'$. If $d^*$ is an original feature, then there cannot be a path containing cuts in features $d_{j,1}, d_{j,2}, \ldots, d_{j,k}$ and in feature $d_{j,\alpha}$ or $d_{j,\beta}$, and thus we get at least $t + 1$ errors. The same argument applies if we make the feature of any cut in the subtree rooted at $z$ be an original feature. Therefore, the cut exchange operations performed in a gadget must essentially just permute the features $d_{j,l}$, which can only affect the classification of the leaf examples belonging to that gadget as other examples have value $\infty$ in those features. As the leaf examples $e_{j,l}^+, e_{j,l}^-$ are initially all correctly classified, there is no possible gain from doing that, and another solution can be obtained by not performing those operations.

Since one can avoid changing any features in original cuts and we apply at most $k - (s' + 1) = k'$ cut exchange operations in the original cuts, the solution can then be mapped back to THRESHOLD ADJUSTMENT by performing the same threshold adjustments in the original cuts. Moreover, by setting the leaf labels to the dominate class, we then obtain a solution for the THRESHOLD ADJUSTMENT instance. □

### A.6 Proof of Proposition 3.7

*Proof.* We can reduce any instance of THRESHOLD ADJUSTMENT to CUT EXCHANGE in polynomial time by Theorem 3.6 such that the size of the decision tree and the number of features in the new instance are bounded by a function (quadratic) of the size of the decision tree in the THRESHOLD ADJUSTMENT instance. The result then follows from THRESHOLD ADJUSTMENT being W[1]-hard with respect to the same parameters by Proposition 3.4. □

## B  Additional Material for Section 4

### B.1  Proof of Theorem 4.1

*Proof.* We adapt the algorithm of Harviainen et al. [14] for subtree raising under the same parameterization. Let $D_i = \{e[i] : e \in E\} \cup \{-\infty\}$ for $i \in [d]$ be the set of distinct values for feature $i$ with an extra value of $-\infty$. A *threshold sequence* $(l_i, r_i)_{i \in [d]}$ encodes for each feature $i \in [d]$ an interval

$(l_i, r_i]$ with $l_i \leq r_i$ and $l_i, r_i \in D_i$, characterizing a subset of the feature space. We thus let

$$E[(l_i, r_i)_{i \in [d]}] := \{e \in E \colon l_i < e[i] \leq r_i \text{ for all } i \in [d]\}$$

be the set of examples within the box defined by a threshold sequence. Let $Q$ be a dynamic programming table whose entry $Q[v, (l_i, r_i)_{i \in [d]}, k']$ is the minimum number of misclassifications achievable if our decision tree is the subtree rooted at $v \in V$ on our initial decision tree $T$ on the set of examples $E[(l_i, r_i)_{i \in [d]}]$ if we make at most $k'$ threshold adjustment operations. In other words, we have a positive answer to THRESHOLD ADJUSTMENT if $Q[r, (-\infty, \infty)_{i \in [d]}, k] \leq t$ when the root of the initial decision tree is $r$. Define $L((l_i, r_i)_{i \in [d]}, j, x)$ as the threshold sequence identical to $(l_i, r_i)_{i \in [d]}$ except that for $i = j$ we have the interval $(\max\{l_i, x\}, r_i)$. Define $R((l_i, r_i)_{i \in [d]}, j, x)$ analogously with an interval $(l_i, \max\{r_i, x\})$.

For a leaf node $v$ and all $k' \leq k$, it suffices to count the number of examples in $E[(l_i, r_i)_{i \in [d]}]$ whose class labels differ from the majority, since the leaf can be relabeled to have the label of the majority. For a cut $v$, there are two cases to consider when combining subresults of the left child $w$ and the right child $u$ of $v$: either we adjust the threshold of $v$ or not. Thus, we get a dynamic programming recurrence

$$Q[v, (l_i, r_i)_{i \in [d]}, k'] = \min \begin{cases} \min_{k'' \in [0, k']} Q[w, L((l_i, r_i)_{i \in [d]}, \mathsf{feat}(v), \mathsf{thr}(v)), k''] \\ \qquad + Q[u, R((l_i, r_i)_{i \in [d]}, \mathsf{feat}(v), \mathsf{thr}(v)), k' - k''], \\ \min_{k'' \in [0, k'-1]} \min_{x \in D_{\mathsf{feat}(v)}} Q[w, L((l_i, r_i)_{i \in [d]}, \mathsf{feat}(v), x), k''] \\ \qquad + Q[u, R((l_i, r_i)_{i \in [d]}, \mathsf{feat}(v), x), k' - k'' - 1]. \end{cases}$$

To recover one of the possible solutions, backtracking in the dynamic programming table is straightforward.

Note that the number of entries is bounded from above by $n \cdot (D+1)^{2d} \cdot k$, and computing the entry for each cut takes $\mathcal{O}(D \cdot k)$ time. For each of the $s$ leaves, the computation takes $\mathcal{O}(n)$ time. Thus, the total time complexity is $\mathcal{O}((D+1)^{2d+1} nk^2 + ns)$. $\qquad\square$

## B.2   Proof of Theorem 4.2

*Proof.* We adapt the algorithm of Harviainen et al. [14] for subtree raising under the same parameterization, proceeding analogously to Theorem 4.1. The only difference is that when we modify a cut, we may also change its associated feature. For a cut $v$ with a left child $w$ and a right child $u$, the dynamic programming recurrence is thus

$$Q[v, (l_i, r_i)_{i \in [d]}, k'] = \min \begin{cases} \min_{k'' \in [0, k']} Q[w, L((l_i, r_i)_{i \in [d]}, \mathsf{feat}(v), \mathsf{thr}(v)), k''] \\ \qquad + Q[u, R((l_i, r_i)_{i \in [d]}, \mathsf{feat}(v), \mathsf{thr}(v)), k' - k''], \\ \min_{k'' \in [0, k'-1]} \min_{j \in [d]} \min_{x \in D_j} Q[w, L((l_i, r_i)_{i \in [d]}, j, x), k''] \\ \qquad + Q[u, R((l_i, r_i)_{i \in [d]}, j, x), k' - k'' - 1]. \end{cases}$$

The time complexity of computing each state increases by a factor of $d$, resulting in the total time complexity of $\mathcal{O}((D+1)^{2d+1} ndk^2 + ns)$. $\qquad\square$

## B.3   Proof of Theorem 4.3

*Proof.* Essentially, we combine the two dynamic programs from Theorems 4.1 and 4.2 while also allowing subtree replacement as a pruning operation. Formally, in DT-PLS we are equipped with numbers $k_{\mathsf{ad}}, k_{\mathsf{ex}}, k_{\mathsf{re}}, k_{\mathsf{ra}}$ which denote the number of allowed threshold adjustments, cut exchanges, subtree replacement operations, and subtree raising operations, respectively. As in Theorem 4.1, we let $D_i = \{e[i] \colon e \in E\} \cup \{-\infty\}$ for $i \in [d]$ be the set of distinct values for feature $i$ with an extra value of $-\infty$. Again, a *threshold sequence* $(l_i, r_i)_{i \in [d]}$ encodes for each feature $i \in [d]$ an interval $(l_i, r_i]$ with $l_i \leq r_i$ and $l_i, r_i \in D_i$, characterizing a subset of the feature space. We thus let

$$E[(l_i, r_i)_{i \in [d]}] := \{e \in E \colon l_i < e[i] \leq r_i \text{ for all } i \in [d]\}$$

be the set of examples within the box defined by a threshold sequence.

Now, however, the definition of table $S$ and the updates are more complicated to incorporate the 4 different operations.

$$Q[v, (l_i, r_i)_{i \in [d]}, k'_{\text{ad}}, k'_{\text{ex}}, k'_{\text{re}}, k'_{\text{ra}}] = \begin{cases} \text{The minimum number of misclassifications achievable} \\ \text{if our decision tree is the subtree rooted at } v \in V \\ \text{on our initial decision tree } T \text{ on the set of} \\ \text{examples } E[(l_i, r_i)_{i \in [d]}] \text{ if we make} \\ \text{(1) at most } k'_{\text{ad}} \text{ threshold adjustments,} \\ \text{(2) at most } k'_{\text{ex}} \text{ cut exchanges,} \\ \text{(3) prune exactly } k'_{\text{re}} \text{ nodes with subtree replacement, and} \\ \text{(4) prune exactly } k'_{\text{ra}} \text{ nodes with subtree raising.} \end{cases}$$

Note that for subtree replacement we demand *exactly* $k'_{\text{re}}$ operations instead of at most since we assume the input decision tree $T'$ is reasonable and thus no subtree replacement operation on $T'$ can reduce the number of errors. We also demand *exactly* $k_{\text{ra}}$ raising operations since usually they increase the number of errors. Thus, we have a positive answer to DT-PLS if $Q[r, (-\infty, \infty)_{i \in [d]}, k_{\text{ad}}, k_{\text{ex}}, k_{\text{re}}, k_{\text{ra}}] \leq t$ when the root of the initial decision tree is $r$. Again, we define $L((l_i, r_i)_{i \in [d]}, j, x)$ as the threshold sequence identical to $(l_i, r_i)_{i \in [d]}$ except that for $i = j$ we have the interval $(\max\{l_i, x\}, r_i)$ and we define $R((l_i, r_i)_{i \in [d]}, j, x)$ analogously with an interval $(l_i, \max\{r_i, x\})$.

Consider a leaf node $v$. We set a corresponding table entry with $k'_{\text{re}} \geq 1$ or $k'_{\text{ra}} \geq 1$ to $\infty$ since in $v$ no subtree replacement operation can be performed. For the remaining table entries corresponding to $v$, that is, $k'_{\text{re}} = 0$ and $k'_{\text{ra}} = 0$, we do the following: For all $k'_{\text{ad}} \leq k_{\text{ad}}$ and $k'_{\text{ex}} \leq k_{\text{ex}}$, it suffices to count the number of examples in $E[(l_i, r_i)_{i \in [d]}]$ whose class labels differ from the majority, since the leaf can be relabeled to have the label of the majority.

For a cut $v$, there are five cases to consider when combining subresults of the left child $w$ and the right child $u$ of $v$: (P1) We adjust the threshold of $v$, (P2) we make a cut exchange, (P3) we perform subtree replacement on $v$, (P4) we perform subtree raising on $v$ and we substitute the subtree rooted at $v$ by the subtree rooted at $u$ or the subtree rooted at $w$ or (P5) $v$ is not changed. Thus, we get a dynamic programming recurrence

$$Q[v, (l_i, r_i)_{i \in [d]}, k'_{\text{ad}}, k'_{\text{ex}}, k'_{\text{re}}, k'_{\text{ra}}] = \min(X_1, X_2, X_3, X_4, X_5)$$

where we always have $k''_{\text{ad}} \in [0, k'_{\text{ad}}]$, $k''_{\text{ex}} \in [0, k'_{\text{ex}}]$, $k''_{\text{re}} \in [0, k'_{\text{re}}]$, and $k''_{\text{ra}} \in [0, k'_{\text{ra}}]$ for

$$X_1 = \min_{\substack{k''_{\text{ad}}, k''_{\text{ex}}, k''_{\text{re}}, k''_{\text{ra}} \\ k''_{\text{ad}} < k'_{\text{ad}}}} \min_{x \in D_{\text{feat}(v)}} Q[w, L((l_i, r_i)_{i \in [d]}, \text{feat}(v), x), k''_{\text{ad}}, k''_{\text{ex}}, k''_{\text{re}}, k''_{\text{ra}}]$$
$$+ Q[u, R((l_i, r_i)_{i \in [d]}, \text{feat}(v), x), k'_{\text{ad}} - k''_{\text{ad}} - 1, k'_{\text{ex}} - k''_{\text{ex}}, k'_{\text{re}} - k''_{\text{re}}, k'_{\text{ra}} - k''_{\text{ra}}],$$

$$X_2 = \min_{\substack{k''_{\text{ad}}, k''_{\text{ex}}, k''_{\text{re}}, k''_{\text{ra}} \\ k''_{\text{ex}} < k'_{\text{ex}}}} \min_{j \in [d]} \min_{x \in D_j} Q[w, L((l_i, r_i)_{i \in [d]}, \text{feat}(v), x), k''_{\text{ad}}, k''_{\text{ex}}, k''_{\text{re}}, k''_{\text{ra}}]$$
$$+ Q[u, R((l_i, r_i)_{i \in [d]}, \text{feat}(v), x), k'_{\text{ad}} - k''_{\text{ad}}, k'_{\text{ex}} - k''_{\text{ex}} - 1, k'_{\text{re}} - k''_{\text{re}}, k'_{\text{ra}} - k''_{\text{ra}}],$$

$$X_3 = \begin{cases} \min(\text{Blue}((l_i, r_i)_{i \in [d]}), \text{Red}((l_i, r_i)_{i \in [d]}))), & \text{if } s_v = k'_{\text{re}} + k'_{\text{ra}} \text{ and } k'_{\text{re}} \geq 1, \\ \infty, & \text{otherwise.} \end{cases}$$

$$X_4 = \min_{\substack{k''_{\text{re}}, k''_{\text{ra}} \\ k''_{\text{ra}} > 0}} \begin{cases} Q[u, (l_i, r_i)_{i \in [d]}, \text{feat}(u), \text{thr}(u), k'_{\text{ad}}, k'_{\text{ex}}, k'_{\text{re}} - k''_{\text{re}}, k'_{\text{ra}} - k''_{\text{ra}}] \text{ if } s_w = k''_{\text{re}} + k''_{\text{ra}} \\ Q[w, (l_i, r_i)_{i \in [d]}, \text{feat}(w), \text{thr}(w), k'_{\text{ad}}, k'_{\text{ex}}, k'_{\text{re}} - k''_{\text{re}}, k'_{\text{ra}} - k''_{\text{ra}}] \text{ if } s_u = k''_{\text{re}} + k''_{\text{ra}}, \end{cases}$$

$$X_5 = \min_{k''_{\mathsf{ad}}, k''_{\mathsf{ex}}, k''_{\mathsf{re}}, k''_{\mathsf{ra}}} Q[w, L((l_i, r_i)_{i \in [d]}, \mathsf{feat}(v), \mathsf{thr}(v)), k''_{\mathsf{ad}}, k''_{\mathsf{ex}}, k''_{\mathsf{re}}, k''_{\mathsf{ra}}]$$
$$+ Q[u, R((l_i, r_i)_{i \in [d]}, \mathsf{feat}(v), \mathsf{thr}(v)), k'_{\mathsf{ad}} - k''_{\mathsf{ad}}, k'_{\mathsf{ex}} - k''_{\mathsf{ex}}, k'_{\mathsf{re}} - k''_{\mathsf{re}}, k'_{\mathsf{ra}} - k''_{\mathsf{ra}}].$$

where $s_v$ is the number of inner nodes in the subtree rooted at $v$, $s_u$ is the number of inner nodes in the subtree rooted at $w$, and $s_w$ is the number of inner nodes in the subtree rooted at $u$, respectively. Moreover, $\mathsf{Blue}((l_i, r_i)_{i \in [d]})$ and $\mathsf{Red}((l_i, r_i)_{i \in [d]})$ are the number of blue and red examples in the box $(l_i, r_i)_{i \in [d]}$, respectively.

Note that in $X_3$ and $X_4$ only node $v$ has to be removed with the respective operation; the remaining nodes in the subtree of $v$, $u$, or $w$ can be removed with any of the two operations.

To recover one of the possible solutions, backtracking in the dynamic programming table is straightforward.

Note that the number of entries is bounded from above by $n \cdot (D + 1)^{2d} \cdot k^4$ where $k = \max(k_{\mathsf{ad}}, k_{\mathsf{ex}}, k_{\mathsf{re}}, k_{\mathsf{ra}})$, and computing the entry for each cut takes $\mathcal{O}(D \cdot k^4)$ time. For each of the $s$ leaves, the computation takes $\mathcal{O}(n)$ time. Thus, the total time complexity is $\mathcal{O}((D + 1)^{2d+1} n k^8 + ns)$. $\qquad\qquad\square$

### B.4   Missing details of Theorem 4.4

We next argue that Algorithm 1 is correct. Clearly, if Algorithm 1 outputs `True` a threshold assignment to each node in $N$ respecting the error vector $\mathbf{t}$ was found and thus the input instance is a yes-instance. For the converse assume that there exists at least one solution. Now we argue that Algorithm 1 outputs `True`. We show this via induction over $|N|$. For the base case $|N| = 0$, no threshold needs to be determined and the algorithm simply checks if the error bound of each leaf is fulfilled. Now, assume an the induction hypothesis that the statement holds for all instances where $|N| < q$. Since the algorithm does nothing for nodes where the threshold is already known, it is safe to assume that the root $r$ of $T$ has an unknown threshold. Moreover, assume that the left subtree $T_\ell$ of $r$ has $x_1$ nodes with an unknown threshold and that the right subtree $T_r$ of $r$ has $x_2$ nodes with an unknown threshold. Clearly, $x_1 < |N|$ and $x_2 < |N|$. Tree $T_r$ witnesses that there is a threshold assignment for all $x_2$ nodes with an unknown threshold in $T_r$ for $E[f_{\mathsf{feat}(n)} > \mathsf{thr}r^*]$, where $\mathsf{thr}(r^*)$ is the threshold of the root $r$ in a fixed solution. An analogous statement is true for $T_\ell$. Thus, in Line 4 we find a threshold $z \geq \mathsf{thr}(p^*)$ because, by the induction hypothesis, the recursive calls to Lines 6 and 7 in which $D[m] \leq \mathsf{thr}(p^*)$ will return `True`. Again, by the induction hypothesis we obtain threshold assignments for all other nodes with an unknown threshold. Consequently, Algorithm 1 is correct.

It remains to analyze the running time of our algorithm: There are $\mathcal{O}(s^s)$ possibilities for the up to $k$ nodes where a threshold adjustment is performed. Moreover, there are $\mathcal{O}(s^t)$ possibilities for the error vector. Finally, it remains to analyze the running time of Algorithm 1. Observe that the number of recursive calls is $\mathcal{O}(\log(D))$. Moreover, the depth is bounded by $s$, the number of inner nodes of $T$, since after each recursive call one additional threshold is fixed. Consequently, Algorithm 1 has a running time of $\log(D)^{\mathcal{O}(s)}$. Analogously to Kobourov et al. [24, Theorem 3.2.], one can show that $\log(D)^{\mathcal{O}(s)} \in \mathcal{O}(s^{2s} \cdot D^{1/s})$. Hence, we obtain the desired running time bound.

### B.5   Proof of Theorem 4.5

*Proof.* We modify the algorithm behind Theorem 4.4 for the FPT-algorithm for $s + t$ for THRESHOLD ADJUSTMENT as follows: After the guess of the $k$ nodes where an operation is performed, we additionally guess which feature is used in any of these $k$ nodes. Note that we need to consider at most $d^k$ possibilities and clearly in one branch this guess is correct. Afterwards, the algorithm behind Theorem 4.4 works completely analogously. Consequently, we obtain an additional factor of $d^k$ in the running time. $\qquad\square$

### B.6   Proof of Proposition 4.6

*Proof.* 1. We enumerate over all subsets of adjusted nodes of size at most $k$ in $\mathcal{O}(s^k)$ time. For each of these subsets, we try all combinations of the new thresholds in time $\mathcal{O}(D^k)$. To each leaf

we assign the most-dominant class label. Finally, we verify whether performing these adjustments results in at most $t$ misclassifications in time $\mathcal{O}(ns)$. In total, we use $\mathcal{O}\left(s^{k+1}D^k n\right)$ time.

2. We use a straightforward brute force algorithm. We enumerate over all $(D+1)^s$ possible combinations of adjustments and test if we made at most $k$ of them. To each leaf we assign the most-dominant class label. Further, we test if we misclassify at most $t$ examples in $\mathcal{O}(sn)$ time on each iteration.

3. We brute force over all subsets of at most $k$ cuts to modify and enumerate over all possible $d$ features $i$ and $D+1$ thresholds in $D_i$. To each leaf we assign the most-dominant class label. Finally, we verify in $\mathcal{O}(ns)$ time whether we get at most $t$ errors. $\qquad\square$

# C  Additional Material for Section 5

## C.1  Proof of Theorem 5.1

*Proof.* Our construction is inspired by a similar construction of Harviainen et al. [14, Thm. 5.10] who showed W[1]-hardness for decision tree pruning with the so-called raising operation for the same parameter.

We reduce from MULTICOLORED INDEPENDENT SET where each color class has the same number $p$ of vertices. An example instance is shown in part $a)$ of Figure 4. Formally, the input is a graph $G$, and $\kappa \in \mathbb{N}$, where the vertex set $V(G)$ of $N$ vertices is partitioned into $V_1, \ldots, V_\kappa$ and $|V_i| = p$ for each $i \in [\kappa]$. More precisely, $V_i := \{v_i^1, v_i^2, \ldots, v_i^p\}$ and $p \cdot \kappa = N$. The question is whether $G$ contains an independent set consisting of exactly one vertex per class $V_i$. MULTICOLORED INDEPENDENT SET is W[1]-hard parameterized by $\kappa$ and cannot be solved in $f(\kappa) \cdot n^{o(\kappa)}$ time unless the ETH fails [3]. The property that all color classes have the same number of vertices is only used to simplify the proof.

**Outline.** For an overview of our reduction, we refer to Figure 4. The idea is to create two features $d_i^<$ and $d_i^>$ per color class $i$ such that all cuts in features $d_i^<$ and $d_i^>$ correspond to a vertex selection in $V_i$. We achieve this as follows: For each pair $d_i^<$ and $d_i^>$ of features we create examples which can only be separated in these two features and which have labels blue (*separating examples*) and red (*choice examples*) alternatingly. Hence, for each possible threshold $x$ in features $d_i^<$ and $d_i^>$, the resulting decision tree $T$ needs to contain either cut $(d_i^<, x)$ or cut $(d_i^>, x)$. Furthermore, for each edge we create a red *edge example*. If vertex $v_i^j \in V_i$ is selected, then all edge examples corresponding to edges having an endpoint in color class $i$ which is *not* $v_i$ will then be correctly classified by the resulting decision tree $T$. Thus, we can only correctly classify an edge example if we *do not* select at least one endpoint of the corresponding edge. Next, we have another feature $d^*$ with only 2 thresholds to ensure that all red choice examples corresponding to selected vertices are correctly classified by the resulting decision tree $T$ and that an edge example gets misclassified as blue if we select both endpoints of the corresponding edge. Finally, the inner nodes of the input decision tree $T'$ are a path with red and blue leaves alternatingly with cuts in a dummy feature $\widehat{d}$. Thus, in order to obtain $T$, all cuts of $T'$ which are in feature $\widehat{d}$ need to be changed to cuts in features $d_i^<$, $d_i^>$, and $d_i^*$, respectively.

**Construction.**

*Description of the data set:* A visualization is shown in part $b)$ of Figure 4.

- For each edge $\{v_i^x, v_j^z\} \in E(G)$ we add an *edge example* $e(v_i^x, v_j^z)$. To all these examples we assign label red.

- For each $i \in [\kappa]$ and each $x \in [p-1]$ we add a blue *separating example* $b(i, x)$.

- For vertex $v_i^x \in V_i$ we create a red *choice example* $c_i^x$.

- We add a red *dummy example* $r_i$ for each $i \in [(p-1) \cdot \kappa + 1]$ and a blue *dummy example* $b_i$ for each $i \in [(p-1) \cdot \kappa]$.

- We create red *enforcing example* $r^*$ and a blue *forcing example* $b^*$. To make the input tree $T'$ reasonable, we add $Z$ copies of $b^*$, where $Z$ is the total number of red examples

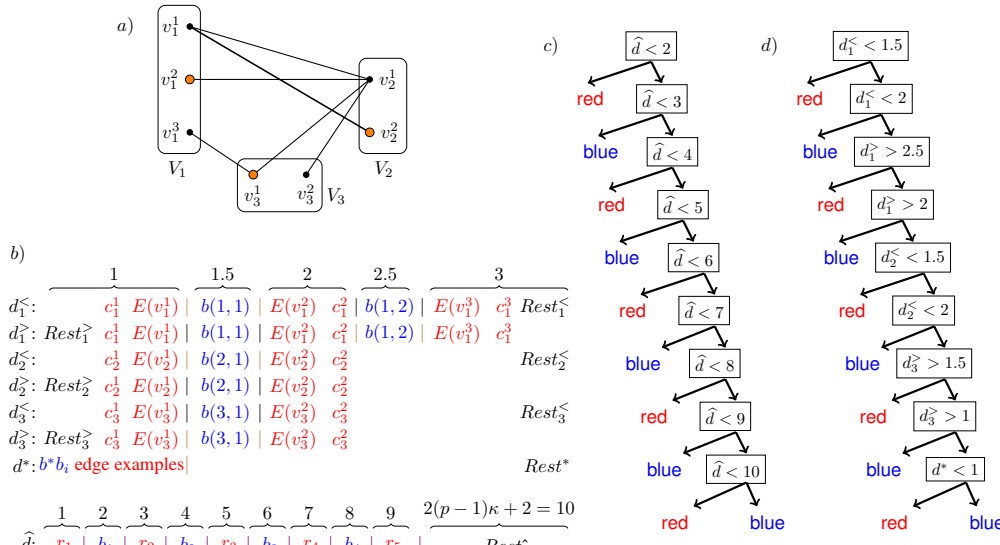

Figure 4: A visualization of the reduction from the proof of Theorem 5.1. Part $a)$ shows a MULTI-COLORED INDEPENDENT SET instance. For the sake of the illustration, the property that all partite sets have the same size is dropped and thus $2(p-1)\kappa + 2 = 10$. A multicolored independent set is depicted in orange. Part $b)$ shows the corresponding classification instance. Here, $E(v_i^j)$ is the set of all edges incident with vertex $v_i^j$. $Rest_i^<$, $Rest_i^>$, $Rest^*$, and $Rest^\wedge$ refers to all other examples not shown in that feature (the precise set differs in each feature and is always a subset of all examples having the default threshold of that feature). All possible cuts in the classification instance are shown by "|". Moreover, cuts in the input decision tree are shown in violet and cuts of the solution decision tree are shown in brown. Part $c)$ shows the input tree $T'$. Part $d)$ shows one possible solution tree $T$.

we create. For simplicity, in the following, we will only talk about the concrete forcing example $b^*$.

Note that we add $M \coloneqq |E(G)|$ edge examples, $N$ choice examples, $N - \kappa = (p-1) \cdot \kappa$ separating examples, $2(p-1) \cdot \kappa + 1$ dummy examples, and 2 further examples. Thus, the number of examples is polynomial in the input size.

For each $i \in [\kappa]$, we add two features $d_i^<$ and $d_i^>$. We also add two other features $d^*$ and $\widehat{d}$. Thus, we have $2 \cdot \kappa + 2$ features.

It remains to describe the coordinates of the examples in the features. Initially, we declare a *default threshold* $\mathsf{default}(d')$ for each feature $d'$. Then, each example $e$ has the default threshold in each feature, unless we assign $e$ a different threshold in that feature.

For each feature $d_i^<$, we set $\mathsf{default}(d_i^<) = p$, for each feature $d_i^>$, we set $\mathsf{default}(d_i^>) = 1$, for feature $d^*$, we set $\mathsf{default}(d^*) = 2$, and for feature $\widehat{d}$, we set $\mathsf{default}(\widehat{d}) = 2(p-1) \cdot \kappa + 2$.

- For each edge example $e = e(v_i^x, v_j^z)$ we set $e[d_i^<] = e[d_i^>] = x$, $e[d_j^<] = e[d_j^>] = z$, and $e[d^*] = 1$. In each other features $e$ is set to the default threshold.

- For the separating example $e = b(i, x)$ we set $e[d_i^<] = e[d_i^>] = x + 1/2$. In each other features $e$ is set to the default threshold.

- For the choice example $e = c_i^x$ we set $e[d_i^<] = e[d_i^>] = x$. In each other features $e$ is set to the default threshold.

- The red enforcing example $r^*$ has the default threshold in every feature. For the blue forcing example $b^*$, we set $b^*[d^*] = 1$, and in each other feature we use the default threshold.

- For the red dummy example $r_i$, we set $r_i[\widehat{d}] = 2i - 1$, and in each other feature we use the default threshold. Finally, for the blue dummy example $b_i$, we set $b_i[\widehat{d}] = 2i$, $b_i[d^*] = 1$, and in each other feature we use the default threshold.

So far, our construction is very similar to the one of Harviainen et al. [14, Thm. 5.10]: Additionally to their construction we have the blue and red dummy examples and the feature $\widehat{d}$. Anything else is identical. The input decision tree, however, is substantially different.

*Description of the input tree $T'$:* For a visualization of $T'$, we refer to part $c)$ of Figure 4. The inner nodes of the input tree $T'$ are a path with cuts only in feature $\widehat{d}$. Moreover, for all cuts the leafs are alternatingly labeled with red and blue, starting with red. Observe that $T$ consists of $(p-1) \cdot 2\kappa + 1$ inner nodes. This completes our construction.

*Parameters $\delta_{\max}$, Error bound $t$ and $\ell$:* Finally, we set $t := 0$, and $k := (p-1) \cdot 2\kappa + 1$. Thus, $\ell = 0$. Note that each edge example differs at most 4 times from the default thresholds, that each separating and each choice example differs exactly 2 times from the default thresholds, that $b^*$ differs exactly once from the default thresholds, that $r^*$ always has the default thresholds, and that each dummy example differs at most 3 times from the default thresholds. Thus, $\delta_{\max} = 6$.

Note that since $\ell = 0$, we can perform a cut exchange on any node and thus the sole purpose of $T'$ is to fix the structure of the inner nodes and the leaf labeling.

*Reasonability of the input tree $T'$:* Observe that each leaf except the blue leaf of the last cut in $T'$ contains exactly one example of the same color. Moreover, observe that this blue leaf of the last cut of $T'$ contains the red enforcing example, all red choice examples, all red edge examples, all blue separation examples, and all $Z$ copies of the blue forcing example, where $Z$ is the total number of red examples, we conclude that this blue leaf contains more blue examples than red examples. Consequently, $T'$ is reasonable.

**Correctness.** We show that $G$ has a multicolored independent set if and only if all cuts of $T'$ can be exchanged to obtain a tree $T$ making at most $t = 0$ errors.

($\Rightarrow$) This direction of the correctness proof is almost analogous to the same direction of the correctness proof of Harviainen et al. [14, Thm. 5.10]. For completeness, we give all details. More precisely, the solution decision tree $T$ in both proofs is identical and here we additionally need to argue that all dummy examples are correctly classified by $T$. For a visualization of $T$, we refer to part $d)$ of Figure 4.

Let $S = \{v_i^{a_i} : i \in [\kappa], a_i \in [p]\}$ be a multicolored independent set of $G$. We perform a cut exchange on all nodes of $T'$ such that in the resulting tree $T$, in feature $d_1^<$ we have exactly one cut at thresholds $x \in \{1.5, 2, 2.5, 3, \ldots, p\}$ for which $x \leq a_i$. Similarly, for each feature $d_i^>$ in tree $T$, we have exactly one cut at thresholds $x \in \{1, 1.5, 2, 2.5, \ldots, p - 1/2\}$ for which $a_i \leq x$. Moreover, in $T$ we have the unique cut in feature $d^*$. Furthermore, in $T$ we first have the cuts in feature $d_1^<$ in ascending order, then the cuts in feature $d_1^>$ in descending order, then in feature $d_2^<$ in ascending order and so on. The last cut of $T$ is in feature $d^*$. In other words, $T$ has the cuts $\{d_1^< < 1.5, d_1^< < 2, \ldots, d_1^< < a_1, d_1^> > p - 1/2, \ldots, d_1^> > a_1, \ldots, d_\kappa^> > a_\kappa, d^* < 1\}$ in that specific order. Moreover, the class assignment to the leaves is not changed.

Since $T'$ has $(p-1) \cdot 2\kappa + 1$ inner nodes and since we perform a cut exchange on each of them, we obtain $k = (p-1) \cdot 2\kappa + 1$ and $\ell = 0$. Thus, it remains to verify that $T$ makes no errors.

*Outline:* First, we make an observation for examples using the default threshold in a feature and second we use this observation to show that all examples are correctly classified by the solution tree $T$.

*Step 1:* Recall that $T$ has no cut in feature $\widehat{d}$. Observe that if any example $e$ lands at some inner node of $T$ corresponding to a cut in feature $d'$ where $d' = d_i^<$ or $d' = d_i^>$ for some $i \in [\kappa]$ and $e$ has the default threshold in that feature $d'$, that is, $e[d'] = \mathsf{default}(d')$, then $e$ will always go to the right subtree of that node. Moreover, any example $e$ with the default threshold in feature $d^*$ is put into the left subtree of the cut in feature $d^*$. Since $T'$ is a path, any example $e$ which has the default threshold in each feature will be contained in the red leaf of the cut $d^* < 1$. Also, in order for an example $e$ to land in a different leaf, we only need to consider cuts of $T$ in features where $e$ has a different threshold than the default threshold.

*Step 2:* We distinguish the different example types.

*Step 2.1:* By construction, the red enforcing example $r^*$ always has the default threshold. Thus $r^*$ ends up in the left child of the last cut of $T$ which is a red leaf. Furthermore, the unique feature in which the blue forcing example $b^*$ does not have the default threshold is $d^*$. Thus, $b^*$ ends up in the right child of the last cut of $T$ which is a blue leaf.

Thus, examples $r^*$ and $b^*$ are correctly classified by $T$.

*Step 2.2:* Consider a blue separating example $e = b(i, z)$. Recall that $z = x + 1/2$ and $x \in [p-1]$ and recall that $a_i \in \mathbb{N}$ is the index of the selected vertex of color class $i$. Without loss of generality, assume that $z < a_i$. By Step 1, $e$ will end up in the cut $d_i^< < 1.5$ of $T$. Also, recall that the next cuts in $T'$ are $d_i^< < 2, \ldots, d_i^< < a_i$ in that specific order. Consequently, $e$ goes to the left subtree of the cut $d_i^< < z + 1/2$, which by construction is a blue leaf. Thus, $e$ is correctly classified as blue by $T$.

*Step 2.3:* Consider a red choice example $e = c_i^x$ where $x \in [p]$.

First, consider the case that $x \neq a_i$. Then the argumentation is almost identical to the blue separating examples: Without loss of generality, assume that $x < a_i$. By Step 1, $e$ will end up in the cut $d_i^< < 1.5$ of $T$. Also, recall that the next cuts in $T$ are $d_i^< < 2, \ldots, d_i^< < a_i$ in that specific order. Consequently, $e$ goes to the left subtree of the cut $d_i^< < x + 1/2$, which by construction is a red leaf.

Second, consider the case that $x = a_i$. Observe that in all cuts of $T$ in features $d_i^<$ and $d_i^>$, example $e$ will always go to the right subtree. Since $e$ has the default threshold in each features different from $d_i^<$ and $d_i^>$, example $e$ ends up in the left leaf of the last cut $d^* < 1$ of $T$ which is a red leaf.

Thus, in both cases $e$ is correctly classified as red by $T$.

*Step 2.4:* Consider a red edge example $e = (v_i^x, v_j^z)$. By assumption, $S$ is a multicolored independent set. Hence, at least one of the two endpoints $v_i^x$ and $v_j^z$ is not contained in $S$. Without loss of generality, assume that $v_i^x \notin S$ and that $i < j$. The argumentation is analog to the red choice examples $c_i^x$ where $x \neq a_i$: Without loss of generality, assume that $x < a_i$. By Step 1, $e$ will end up in the cut $d_i^< < 1.5$ of $T$. Also, recall that the next cuts in $T$ are $d_i^< < 2, \ldots, d_i^< < a_i$ in that specific order. Consequently, $e$ goes to the left subtree of the cut $d_i^< < x + 1/2$, which by construction is a red leaf. Thus, $e$ is correctly classified as red by $T$.

*Step 2.5:* It remains to consider the dummy examples. First, consider a red dummy example $r_h$. Recall that $r_h$ uses the dummy thresholds in all features except $\widehat{d}$. Since $\widehat{d}$ is not used in $T$, the red dummy example $r_h$, analogously to the red enforcing example $r^*$, ends up in the left child of the last cut of $T$ (which is $d^* < 1$) which is a red leaf. Second, consider a blue dummy example $b_h$. Recall that $b_h$ uses the dummy thresholds in all features except $\widehat{d}$ and $d^*$. Hence, analogously to the blue forcing example $b^*$, the blue dummy example $b_h$, ends up in the right child of the last cut of $T$ (which is $d^* < 1$) which is a blue leaf.

Consequently, the tree $T$ has no classification errors.

($\Leftarrow$) Let $T$ be a solution of the cut exchange problem, that is, up to all cuts of the initial tree $T'$ have been changed such that $T$ makes no errors.

*Outline:* This direction of the correctness proof follows a similar route as the corresponding direction of the correctness proof of Harviainen et al. [14, Thm. 5.10]. Step 4, however, is substantially different.

We first show that $T$ has to include the unique cut in feature $d^*$. Second, we show that $T$ needs to contain at least on of the two cuts $d_i^< < x$ and $d_i^> > x - 1/2$ for each $i$ and each $x$. Third, because of our choice of the size of the initial tree $T'$ and $\ell = 0$ we then conclude that for each $i$ and each $x$ exactly one of the cuts $d_i^< < x$ and $d_i^> > x - 1/2$ has to be preserved. Fourth, we show that the cuts of $T$ in a feature $d_i^<$ (or $d_i^>$) do not have *gaps*, that is, if $x$ is the largest (smallest) threshold, such that the cut $d_i^< < x$ ($d_i^> > x$) is included in $T$, then also all cuts $d_i^< < z$ for each $z < x$ ($d_i^> > z$ for each $x < z$) have to be contained in $T$. For example, the tree shown in part $d$) of Figure 4 which has 0 errors fulfills this property. Fifth, we use this solution structure to identify a selected vertex of each color class. Let $S$ be the corresponding vertex set. Finally, we show that $S$ has to be a multicolored independent set.

*Step 1:* Note that the blue forcing example $b^*$ and that the red enforcing example $r^*$ only differ in the binary feature $d^*$. Thus, $T$ has to contain the cut $d^* < 1$.

*Step 2:* Our aim is to show that at least one of the cuts $d_i^< < x$ and $d_i^> > x - 1/2$ for any $i \in [\kappa]$ and $x \in \{1.5, 2, 2.5, \ldots, p\}$ has to be included in $T$. Without loss of generality assume that $x$ is an integer. Note that $x \geq 2$. By construction, for the blue separating example $e = b(i, x - 1)$ we have $e[d_i^<] = e[d_i^>] = x - 1/2$ and for the red choice example $e = c_i^x$ we have $e[d_i^<] = e[d_i^>] = x$. Furthermore, note that $b(i, x-1)$ and $c_i^x$ have the default threshold in each other feature. Consequently, only the cuts $d_i^< < x$ and $d_i^> > x - 1/2$ can separate $b(i, x-1)$ and $c_i^x$. Since $T$ has no classification errors, we thus conclude that at least one of the cuts $d_i^< < x$ and $d_i^> > x - 1/2$ has to be included in $T$.

*Step 3:* Recall that $s = (p-1) \cdot 2\kappa + 1$ and $\ell = 0$. By Step 1, $T$ has to include the cut $d^* > 1$. By Step 2, at least one of the cuts $d_i^< < x$ and $d_i^> > x - 1/2$ for any $i \in [\kappa]$ and $x \in \{1.5, 2, 2.5, \ldots, p\}$ has to be contained in $T$. Note that these are exactly $(p-1) \cdot 2\kappa$ pairs of distinct cuts. Consequently, $T$ has to contain exactly one of the cuts $d_i^< < x$ and $d_i^> > x - 1/2$.

*Step 4:* We show that for each $i \in [\kappa]$ there is a threshold $x_i$ such that $T$ contains all cuts $d_i^< < z$ where $z \leq x_i$ and all cuts $d_i^> > z$ where $z > x_i$. Assume towards a contradiction that this is not true. Let $P_i^< \subseteq \{1.5, 2, 2.5, \ldots, p\}$ be the subset of consecutive thresholds including 1.5 such that for each $x_i \in P_i^<$ the tree $T$ contains the cut $d_i^< < x_i$. Analogously, let $P_i^> \subseteq \{1.5, 2, 2.5, \ldots, p\}$ be the subset of consecutive thresholds including $p$ such that for each $x_i \in P_i^>$ the tree $T$ contains the cut $d_i^> > x_i - 1/2$. According to our assumption, $P_i^< \cup P_i^> \neq \{1.5, 2, 2.5, \ldots, p\}$. Now consider the first cut $d_i^< < x$ or $d_i^> > x - 1/2$ of $T$ (from the root) such that $x \in \{1.5, 2, 2.5, \ldots, p\} \setminus (P_i^< \cup P_i^>)$. Without loss of generality assume it is the cut $d_i^< < x$. By our choice, $x \notin P_i^<$. Let $z$ be the largest threshold in $P_i^<$. Clearly, $z < x$. Moreover, $z \leq x - 1$, because otherwise if $z = x - 1/2$ then we would have $x \in P_i^<$, a contradiction to the fact that $P_i^<$ is maximal. Also, observe that for threshold $x - 1/2$ we have $x - 1/2 \notin P_i^< \cup P_i^>$. Without loss of generality, we assume that $x$ is an integer. We now exploit the fact that $x \notin P_i^< \cup P_i^>$ is the first threshold such that either $d_i^< < x$ or $d_i^> > x - 1/2$ is contained in $T$. Since also $x - 1/2 \notin P_i^< \cup P_i^>$, and according to our choice of $x$ at the time when in $T$ cut $d_i^< < x$ is considered, neither the cut $d_i^< < x - 1/2$ nor the cut $d_i^> > x - 1$ was considered. Thus, the red choice example $c_i^{x-1}$ and the blue separating example $b(i, x - 1)$ are both moved into the left subtree of the node with cut $d_i^< < x$ which is a leaf implying that one of them gets misclassified, a contradiction to the fact that $t = 0$. Thus, $P_i^<$ and $P_i^>$ form a partition of $\{1.5, 2, 2.5, \ldots, p\}$.

We would like to note that one can additionally show that all cuts $d_i^< < x$ contained in $T$ appear in ascending order in $T$, that is no cut $d_i^< < z$ appears before a cut $d_i^< < x$ for some $z > x$. This fact, however, is not necessary to obtain the multicolored independent set.

*Step 5:* Consider one fixed $i \in [\kappa]$. Let $x_i$ be the largest threshold such that the cut $d_i^< < x_i$ is contained in in $T$. In other words, $x_i$ is the largest threshold contained in $P_i^<$. Thus, each threshold which is smaller than $x$ is also contained in $P_i^<$ and all thresholds larger than $x_i$ are contained in $P_i^>$. Next, assume towards a contradiction that $x$ is no integer, that is, $x_i = q + 1/2$ for some integer $q \in [p-1]$. Now, observe that the blue separating example $b(i, q)$ is put in the right subtree of each cut $d_i^< < z$ for each $z \leq x_i$ and for each cut $d_i^> > z$ for each $x_i \leq z$. Recall that $d_i^<$ and $d_i^>$ are the only features such that $b(i, q)$ does not use the default thresholds. Now, since the red enforcing example $r^*$ always uses the default thresholds, we conclude that $T$ cannot distinguish $b(i, q)$ and $r^*$ and consequently $T$ makes at least one error, a contradiction to the fact that $t = 0$. Hence, $x_i$ is an integer.

We let $v_i^{x_i}$ be the selected vertex of color class $i$. Furthermore, let $S := \{v_i^{x_i} : i \in [\kappa]\}$.

*Step 6:* It remains to verify that $S$ is a multicolored independent set. By definition, $S$ contains exactly one vertex of each color class. Hence, it remains to show that $S$ is an independent set.

Observe that since $T$ has no classification errors, it is sufficient to show that $T$ cannot distinguish the blue forcing example $b^*$ and an red edge example $e = (v_i^x, v_j^z)$ if both endpoints $v_i^x$ and $v_j^z$ are contained in $S$: Let $e = e(v_i^x, v_j^z)$ be an edge example where $v_i^x, v_j^z \in S$. Note that analogously to Step 5, the red edge example $e$ ends up in the right subtree of each cut in features $d_i^<, d_i^>, d_j^<$, and $d_j^>$. Since the blue forcing example $b^*$ uses the default thresholds in these 4 features, this is also true for $b^*$. Now, observe that the only other feature where $e$ or $b^*$ to not use the default thresholds

is $d^*$. But in $d^*$ we have $e[d^*] = 1 = b^*[d^*]$. Thus, $T$ cannot distinguish the blue forcing example $b^*$ and the red edge example $e = (v_i^x, v_j^z)$. Consequently, $S$ is also an independent set.

**Lower Bound.** Recall that $d = 2 \cdot \kappa + 2, \delta_{\max} = 6, \ell = 0$, and $t = 0$. Since MULTICOLORED INDEPENDENT SET is W[1]-hard with respect to $\kappa$ [3], we obtain that CUT EXCHANGE is W[1]-hard with respect to $d$ even if $\delta_{\max} = 6, \ell = 0$, and $t = 0$. Furthermore, since MULTICOLORED INDEPENDENT SET cannot be solved in $f(\kappa) \cdot n^{o(\kappa)}$ time unless the ETH fails [3], we observe that CUT EXCHANGE cannot be solved in $f(d) \cdot |\mathcal{I}|^{o(d)}$ time if the ETH is true, where $|\mathcal{I}|$ is the overall instance size, even if $\delta_{\max} = 6, \ell = 0$, and $t = 0$. $\qquad\square$

## C.2   Proof of Theorem 5.2

*Proof.* Our construction is inspired by a similar construction of Harviainen et al. [14, Thm. 5.10] who showed W[1]-hardness for decision tree pruning with the so-called raising operation for the same parameter. Our construction, however, is substantially more involved due to the restrictive nature of the threshold adjustment operation.

We reduce from MULTICOLORED INDEPENDENT SET where each color class has the same number $p$ of vertices. An example instance is shown in part $a)$ of Figure 5. Formally, the input is a graph $G$, and $\kappa \in \mathbb{N}$, where the vertex set $V(G)$ of $N$ vertices is partitioned into $V_1, \ldots, V_\kappa$ and $|V_i| = p$ for each $i \in [\kappa]$. More precisely, $V_i := \{v_i^1, v_i^2, \ldots, v_i^p\}$ and $p \cdot \kappa = N$. The question is whether $G$ contains an independent set consisting of exactly one vertex per class $V_i$. MULTICOLORED INDEPENDENT SET is W[1]-hard parameterized by $\kappa$ and cannot be solved in $f(\kappa) \cdot n^{o(\kappa)}$ time unless the ETH fails [3]. The property that all color classes have the same number of vertices is only used to simplify the proof.

**Outline.** For an overview of our reduction, we refer to Figure 5. Similar to Theorem 5.1, the idea is to create two features $d_i^<$ and $d_i^>$ per color class $i$ such that all cuts in features $d_i^<$ and $d_i^>$ correspond to a vertex selection in $V_i$. We achieve this as follows: For each pair $d_i^<$ and $d_i^>$ of features we create examples which can only be separated in these two features and which have labels blue (*separating examples*) and red (*choice examples*) alternatingly. These are the *important* thresholds of features $d_i^<$ and $d_i^>$. Hence, for each possible threshold $x$ in features $d_i^<$ and $d_i^>$, the resulting decision tree $T$ needs to contain either cut $(d_i^<, x)$ or cut $(d_i^>, x)$. Furthermore, for each edge we create a red *edge example*. If vertex $v_i^j \in V_i$ is selected, then all edge examples corresponding to edges having an endpoint in color class $i$ which is *not* $v_i$ will then be correctly classified by the resulting decision tree $T$. Thus, we can only correctly classify an edge example if we *do not* select at least one endpoint of the corresponding edge. Next, we have another feature $d^*$ with only 2 thresholds to ensure that all red choice examples corresponding to selected vertices are correctly classified by the resulting decision tree $T$ and that an edge example gets misclassified as blue if we select both endpoints of the corresponding edge.

In the proof in Theorem 5.1 for cut exchange, the initial tree $T'$ was a path with red and blue leaves alternatingly. For the threshold adjustment operation we need a more complex structure of $T'$: Since we are only allowed to do threshold adjustments, we cannot add a dummy feature with dummy cuts as in the proof in Theorem 5.1 for cut exchange. Instead, we extend each feature $d_i^<$ and $d_i^>$ by *dummy thresholds* which are used in the initial tree $T'$. Since $T'$ is reasonable, we also add new *dummy examples* $r_i^x$ and $b_i^x$ alternatingly in features $d_i^<$ and $d_i^>$. This, however, creates a new problem: since $t = 0$ doing a threshold adjustment in feature $d_i^<$ moves both a red and a blue dummy example in the same (left) subtree. Thus, as in the proof in Theorem 5.1 the left subtree is not simply a leaf, but is another inner node with a cut in a new *rescue feature* $p_i^<$ to separate the differently labeled dummy examples. Our construction of these rescue features together with the tightness of the budget $k$ ensures that in any optimal solution no threshold adjustment in any rescue feature is possible. In other words, all threshold adjustments need to be done within the features $d_i^<$ and $d_i^>$.

**Construction.** Compared to Harviainen et al. [14, Thm. 5.10] and to our reduction in Theorem 5.1, our construction here is significantly more involved. On the one hand, we need way more examples and features and on the other hand, the input decision tree is not only a path. Next, we describe our construction in detail.

*Description of the data set:* A visualization is shown in part $d)$ of Figure 5.

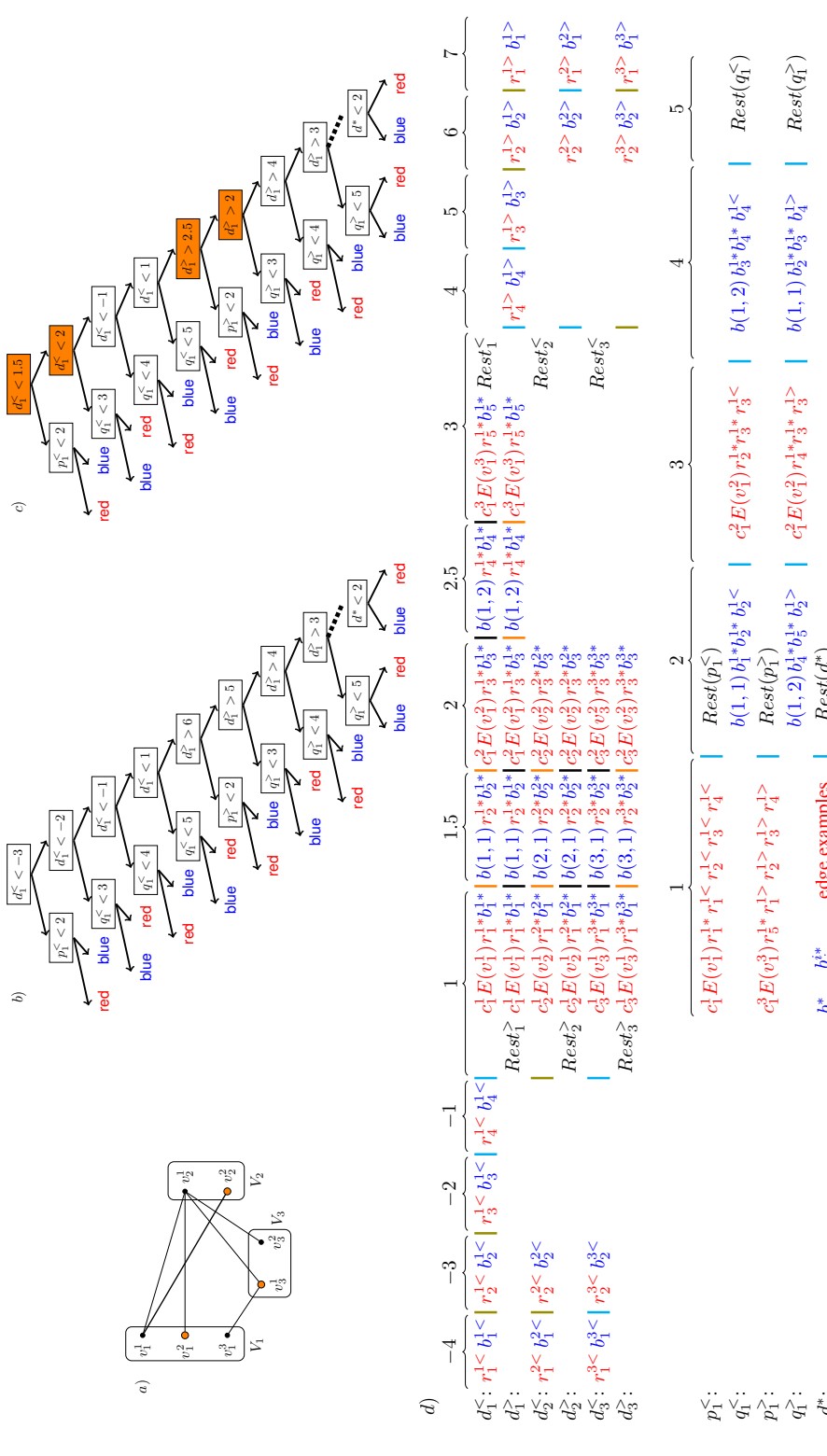

Figure 5: A visualization of the reduction from the proof of Theorem 5.2. Part $a$) shows a MULTICOLORED INDEPENDENT SET instance. For the sake of the illustration, the property that all partite sets have the same size is dropped. A multicolored independent set is depicted in orange. The vertex selection in the partite set $V_1$, and the final cut in feature $d^*$. Part $b$) shows the part of the input tree $T'$ corresponding to the vertex selection in the partite set $V_1$ together with the final cut in feature $d^*$. Part $c$) shows the part of a solution tree $T$ corresponding to the vertex selection in the partite set $V_1$ together with the final cut in feature $d^*$. The cuts where a threshold adjustment was applied are marked orange and they indicate the selection of vertex $v_1^2 \in V_1$. Part $d$) shows the corresponding classification instance. Here, $E(v_i^j)$ is the set of all edges incident with vertex $v_i^j$. $Rest$ refers to all other examples not shown in that feature (the precise sets differs in each feature and is always a subset of all examples having the default threshold of that feature). Features $x_j^<$ and $x_j^>$ for $x \in \{p, q\}$ and $j \in \{2, 3\}$ are not shown. Note that thresholds $-4$ to $-1$ only exist for features $d_i^<$ and that thresholds $d_i^>$ only exist for features $d_i^>$. All possible cuts in the classification instance are shown by "|". Moreover, we use a 4 coloring to distinguish cuts of the initial tree $T'$ and a solution tree $T$: 1) cyan cuts are used by both $T'$ and $T$, 2) brown cuts are only used by $T'$, orange cuts are only used by $T$, and 4) black cuts are neither used by $T'$ nor $T$.

- For each edge $\{v_i^x, v_j^z\} \in E(G)$ we add an *edge example* $e(v_i^x, v_j^z)$. To all these examples we assign label red.

- For each $i \in [\kappa]$ and each $x \in [p-1]$ we add a blue *separating example* $b(i, x)$.

- For vertex $v_i^x \in V_i$ we create a red *choice example* $c_i^x$.

- We create red *enforcing example* $r^*$ and a blue *forcing example* $b^*$. To make the input tree $T'$ reasonable, we add $(N^2 + 1)$ copies of $b^*$. For simplicity, in the following, we will only talk about the concrete forcing example $b^*$.

- For each feature $d_i^<$ and each $j \in [2(p-1)]$ we add a blue *dummy example* $b_j^{i<}$ and a red *dummy example* $r_j^{i<}$. Analogously, for each feature $d_i^>$ and each $j \in [2(p-1)]$ we add a blue *dummy example* $b_j^{i>}$ and a red *dummy example* $r_j^{i>}$.

- For each $i \in [k]$ and each $j \in [2(p-1)+1]$, we add a blue *filler example* $b_j^{i*}$ and a red *filler example* $r_j^{i*}$.

Note that we add $M := |E(G)|$ edge examples, $N$ choice examples, $N - \kappa = (p-1) \cdot \kappa$ separating examples, $8(p-1) \cdot \kappa$ dummy examples, $(4(p-1)+2) \cdot \kappa$ filler examples, and 2 further examples. Thus, the number of examples is polynomial in the input size.

For each $i \in [\kappa]$, we add six features $d_i^<, d_i^>, p_i^<, p_i^>, q_i^<,$ and $q_i^>$. We also add another feature $d^*$. Thus, we have $6 \cdot \kappa + 1$ features.

It remains to describe the coordinates of the examples in the features. Initially, we declare a *default threshold* $\mathsf{default}(d')$ for each feature $d'$. Then, each example $e$ has the default threshold in each feature, unless we assign $e$ a different threshold in that feature.

- For each feature $d_i^<$, we set $\mathsf{default}(d_i^<) = p$, for each feature $d_i^>$, we set $\mathsf{default}(d_i^>) = 1$.

- For each feature $p_i^<$, we set $\mathsf{default}(p_i^<) = 2$. Analogously, for each feature $p_i^>$, we set $\mathsf{default}(p_i^>) = 2$.

- For each feature $q_i^<$, we set $\mathsf{default}(q_i^<) = 2(p-1)+1$. Analogously, for each feature $q_i^>$, we set $\mathsf{default}(q_i^>) = 2(p-1)+1$.

- For feature $d^*$, we set $\mathsf{default}(d^*) = 2$.

Now, we are ready to assign the coordinates to all examples.

- For each edge example $e = e(v_i^x, v_j^z)$ we set $e[d_i^<] = e[d_i^>] = x$, $e[d_j^<] = e[d_j^>] = z$, and $e[d^*] = 1$. Next, we distinguish three different values of $x$: (1) if $x = 1$, we set $e[p_i^<] = 1$. (2) if $x = p$, we set $e[p_i^>] = 1$. (3) if $1 < x < p$, we set $e[q_i^<] = 2x-1$ and $e[q_i^<] = 2(p-x)+1$. We do similar for $z$, that is, (1) if $z = 1$, we set $e[p_j^<] = 1$. (2) if $z = p$, we set $e[p_j^>] = 1$. (3) if $1 < z < p$, we set $e[q_j^<] = 2z-1$ and $e[q_j^<] = 2(p-z)+1$. In each other features $e$ is set to the default threshold.

- For the separating example $e = b(i, x)$ we set $e[d_i^<] = e[d_i^>] = x + 1/2$. Moreover, we set $e[p_i^<] = 2x$, and $e[p_i^>] = 2(p-x)$. In each other features $e$ is set to the default threshold.

- For the choice example $e = c_i^x$ we set $e[d_i^<] = e[d_i^>] = x$. Next, we distinguish three different values of $x$: (1) if $x = 1$, we set $e[p_i^<] = 1$. (2) if $x = p$, we set $e[p_i^>] = 1$. (3) if $1 < x < p$, we set $e[q_i^<] = 2x - 1$ and $e[q_i^<] = 2(p-x)+1$. In each other features $e$ is set to the default threshold.

- The red enforcing example $r^*$ has the default threshold in every feature. For the blue forcing example $b^*$, we set $b^*[d^*] = 1$, and in each other feature we use the default threshold.

- Let $i \in [\kappa]$ and $j \in [2(p-1)]$.

(A) For the blue dummy example $e = b_j^{i<}$, we set $e[d_1^\leq] = -2(p-1)-1+j = -2p+1+j$. Note that $-2(p-1) \leq e[d_1^\leq] \leq -1$. Moreover, if $j$ is even, we set $e[q_i^\leq] = j$.

(B) For the blue dummy example $e = b_j^{i>}$, we set $e[d_1^>] = p + 2(p-1) - j = 3p - 1 - j$. Note that $p + 1 \leq e[d_1^\leq] \leq 3p - 2$. Moreover, if $j$ is even, we set $e[q_i^>] = j$.

(C) For the red dummy example $e = r_j^{i<}$, we set $e[d_1^\leq] = -2(p-1)-1+j = -2p+1+j$. Note that $-2(p-1) \leq e[d_1^\leq] \leq -1$. Also, we set $e[p_i^\leq] = 1$. Moreover, if $j > 1$ and $j$ is odd, we set $e[q_i^\leq] = j$.

(D) For the red dummy example $e = r_j^{i>}$, we set $e[d_1^>] = p + 2(p-1) - j = 3p - 1 - j$. Note that $p + 1 \leq e[d_1^\leq] \leq 3p - 2$. Also, we set $e[p_i^>] = 1$. Moreover, if $j > 1$ and $j$ is odd, we set $e[q_i^>] = j$.

- Let $i \in [\kappa]$ and $j \in [2p - 1]$.

  (A) For the blue filler example $e = b_j^{i*}$, we set $e[d_i^\leq] = (j+1)/2 = e[d_i^>]$, and $e[d^*] = 1$. Next, we distinguish the precise value of $j$: (1) if $j = 1$, we set $e[q_i^\leq] = 2$. (2) if $j = 2p - 1$, we set $e[q_i^>] = 2$. (3) if $1 < j < 2p - 1$ and $j$ is odd, we set $e[q_i^\leq] = j + 1$ and $e[q_i^>] = 2p + 1 - j$. (4) otherwise, if $j$ is even, we set $e[q_i^\leq] = j$, and $e[q_i^>] = 2p - j$.

  (B) For the red filler example $e = r_j^{i*}$, we set $e[d_i^\leq] = (j+1)/2 = e[d_i^>]$. Next, we distinguish the precise value of $j$: (1) if $j = 1$, we set $e[p_i^\leq] = 1$. (2) if $j = 2p - 1$, we set $e[p_i^>] = 1$. (3) if $j = 2$, we set $e[q_i^\leq] = 3$. (4) if $1 < j < 2p - 1$ and $j$ is odd, we set $e[q_i^\leq] = j = e[q_i^>]$. (5) otherwise, if $2 < j$ and $j$ is even, we set $e[q_i^\leq] = j + 1$, and $e[q_i^>] = j - 1$.

*Description of the input tree $T'$:* For a visualization of parts of $T'$, we refer to part $b)$ of Figure 5. Intuitively, tree $T'$ consists of a path $P$ of cuts in features $d_i^\leq$, $d_i^>$, and $d^*$, and each left subtree of this path (except for the unique cut in feature $d^*$) consists of a unique cut in features $x_i^\leq$ or $x_i^>$ where $x \in \{p, q\}$.

First, we formally describe path $P$: It consists of the cuts $d_i^\leq < -2p + 3, d_i^\leq < -2p + 4, \ldots, d_i^\leq < -1, d_i^\leq < 1, d_i^> > 3p - 3, \ldots, d_i^> > p, d_2^\leq < -2p + 3, \ldots, d_\kappa^> > p, d^* < 2$. Second, we describe the remaining cuts: (1) The left subtree of the cut $d_i^\leq < -2p + 3$ consists of the cut $p_i^\leq < 2$. (2) The left subtree of the cut $d_i^> > 3p - 3$ consists of the cut $p_i^> < 2$. (3) The left subtree of the cut $d_i^\leq < -2p + 2 + j$ for $j \geq 2$ consists of the cut $q_i^\leq < j + 1$. (4) The left subtree of the cut $d_i^> > -3p - 2 - j$ for $j \geq 2$ consists of the cut $q_i^> < j + 1$.

Finally, it remains to describe the class assignment to the leaves: The left leaf of the cuts $p_i^\leq < 2$, $p_i^> < 2$, $q_i^\leq < j$, and $q_i^> < j$ for even $j$ is red and their right child is blue. For all remaining cuts, that is for cuts $q_i^\leq < j$, $q_i^> < j$ for odd $j$, and for $d^* < 2$, the left child is blue and the right child is red.

Observe that $T'$ consists of exactly $8(p-1) \cdot \kappa + 1$ cuts.

*Parameters $\delta_{\max}$, Error bound $t$ and $\ell$:* Finally, we set $t := 0$, and $k := (p-1) \cdot 2\kappa$. Note that each edge example differs at most 13 times from the default thresholds, that each separating and each choice example differs exactly 6 times from the default thresholds, that $b^*$ differs exactly once from the default thresholds, that $r^*$ always has the default thresholds, that each dummy example differs at most 3 times from the default thresholds, and that each filler example differs 7 times from the default thresholds. Thus, $\delta_{\max} \leq 20$.

*Reasonability of input tree $T'$:* Observe that each leaf attached to a cut in feature $x_i^\leq$ or $x_i^>$ for $x \in \{p, q\}$ and $i \in [\kappa]$ contains exactly one dummy example of the same class. Moreover, observe that the red leaf of the cut $d^* < 2$ contains all blue separating example, all red choice examples, all red filler examples, and the red enforcing example. Thus, the red leaf contains more red examples than blue examples. Finally, consider the blue leaf of the cut $d^* < 2$. This leaf contains all red edge examples, all blue filler examples, and the $(N^2 + 1)$ copies of the blue forcing example. Thus, the blue leaf contains more blue examples than red examples. Consequently, the input tree $T'$ is reasonable.

**Correctness.** We show that $G$ has a multicolored independent set if and only if on $k = (p-1)\cdot 2\kappa$ cuts of $T'$ we can perform a threshold adjustment operation to obtain a tree $T$ making at most $t = 0$ errors.

($\Rightarrow$) Let $S = \{v_i^{a_i} : i \in [\kappa], a_i \in [p]\}$ be a multicolored independent set of $G$. We now show how to construct a tree $T$ from $T'$ with at most $k = (p-1)\cdot 2\kappa$ thresholds adjustments such that $T$ has $0$ errors. For a visualization of parts of $T$, we refer to part $c)$ of Figure 5.

For each color class $i$, we perform a threshold adjustment operation in the first $2(a_i - 1)$ cuts of $T'$ in feature $d_i^<$ and in the first $2(p - a_i)$ cuts in feature $d_i^>$. Moreover, the newly adjusted thresholds in $d_i^<$ range from $1.5$ to $a_i$ in that order and the newly adjusted thresholds in $d_i^>$ range from $p - 1/2$ to $a_i$ in that order. Note that these are $2(p-1)$ threshold adjustment operations per color class and thus $k = (p-1)\cdot 2\kappa$ threshold adjustment operations in total. Now, we describe these operations in detail: We change cut $d_i^< < -2(p-1) + 1 = -2p + 3$ to $d_i^< < 1.5$, $d_i^< < -2p + 4$ to $d_i^< < 2$, and so on, until $d_i^< < -2(p-1) + 2(a_i - 1) = -2(p - a_i)$ to $d_i^< < a_i$. Moreover, we change cut $d_i^> > 3p - 3$ to $d_i^> > p - 1/2$, $d_i^> > 3p - 4$ to $d_1^> > p - 1$, and so on, until $d_i^> > 3p - 2 - 2(p - a_i) = p + 2a_i - 2$ to $d_i^> > a_i$. Moreover, note that we do not change the class of any leaf.

Observe that the new tree $T$ might not be reasonable anymore: For example, consider the selection of vertex $v_1^2 \in V_1$ shown in part $c)$ of Figure 5: The cut $d_1^< < -1$ puts all examples into its right subtree since its parent has the stronger cut $d_i^< < 2$.

Now, it remains to verify that $T$ makes no errors.

*Outline:* First, we make an observation for examples using the default threshold in a feature and second we use this observation to show that all examples are correctly classified by the solution tree $T$.

*Step 1:* Observe that if any example $e$ lands at some inner node of $T$ corresponding to a cut in feature $d'$ where $d' = d_i^<$ or $d' = d_i^>$ for some $i \in [\kappa]$ and $e$ has the default threshold in that feature $d'$, that is, $e[d'] = \mathsf{default}(d')$, then $e$ will always go to the right subtree of that node. Moreover, any example $e$ with the default threshold in feature $d^*$ is put into the right subtree of the cut in feature $d^*$ which is a *lneg* leaf. Consequently, any example $e$ which has the default threshold in each feature will be contained in the red leaf of the cut $d^* < 2$. Also, in order for an example $e$ to land in a different leaf, we only need to consider cuts of $T$ in features where $e$ has a different threshold than the default threshold.

*Step 2:* We distinguish the different example types and show that $T$ makes no errors.

*Step 2.1:* By construction, the red enforcing example $r^*$ always has the default threshold. Thus $r^*$ ends up in the right child of the last cut of $T$ which is a red leaf. Furthermore, the unique feature in which the blue forcing example $b^*$ does not have the default threshold is $d^*$. Thus, $b^*$ ends up in the left child of the last cut of $T$ which is a blue leaf.

Thus, examples $r^*$ and $b^*$ are correctly classified by $T$.

*Step 2.2:* Consider a blue separating example $e = b(i, z)$. Recall that $z = x + 1/2$ and $x \in [p - 1]$ and recall that $a_i \in \mathbb{N}$ is the index of the selected vertex of color class $i$.

First we consider the case $z \le a_i$. This implies that at least one threshold adjustment is done in feature $d_i^<$. By Step 1, $e$ will end up in the cut $d_i^< < 1.5$ of $T$. Afterwards, the next cuts are $d_i^< < 2, d_i^< < 2.5, \ldots, d_i^< < a_i$ in that specific order. Consequently, $e$ goes to the left subtree of the cut $d_i^< < z + 1/2$ (note that $z + 1/2$ is an integer and $z + 1/2 \ge 2$). Observe that the root of this left subtree is the cut $q_i^< < h$ for some $h \le 2p - 1$ such that $h$ is odd. Consequently, $e$ is put into the left subtree of the cut $q_i^< < h$ which is a blue leaf. Thus, $e$ is correctly classified as blue by $T$.

Second, we consider the case $z > a_i$. Observe that all cuts in feature $d_i^<$ have the form $d_i^< < h$ with some threshold $h \le a_i < z$. Hence, $e$ is put into the right subtree of each cut in feature $d_i^<$ of $T$. Now, the argumentation is analogously as in the first case for feature $d_i^>$ and since $T$ contains the cut $d_i^> > a_i$, example $e$ is put into the left subtree of one of the cuts in feature $d_i^>$.

*Step 2.3:* Consider a red choice example $e = c_i^x$ where $x \in [p]$.

First, consider the case that $x \ne a_i$. Then the argumentation is almost identical to the blue separating examples:

(A) assume that $x < a_i$. By Step 1, $e$ will end up in the cut $d_i^< < 1.5$ of $T$. Also, recall that the next cuts in $T$ are $d_i^< < 2, d_i^< < 2.5, \ldots, d_i^< < a_i$ in that specific order. Consequently, $e$ goes to the left subtree of the cut $d_i^< < x + 1/2$. If, $x = 1$, then the root of this left subtree is the cut $p_i^< < 2$. Otherwise, if $x \geq 2$, then the root of this left subtree is the cut $q_i^< < h$ for some $h \leq 2p - 1$ such that $h$ is even. Observe that in both cases $e$ is put into the left subtree of the cut $p_i^< < 2$ or $q_i^< < h$ which is a red leaf.

(B) assume that $x > a_i$. Observe that all cuts in feature $d_i^<$ have the form $d_i^< < h$ with some threshold $h \leq a_i < x$. Hence, $e$ is put into the right subtree of each cut in feature $d_i^<$ of $T$. Now, the argumentation is analogously as in the first case for feature $d_i^>$ and since $T$ contains the cut $d_i^> > a_i$, example $e$ is put into the left subtree of one of the cuts in feature $d_i^>$.

Second, consider the case that $x = a_i$. Note that in $T$ all cuts in feature $d_i^<$ have the form $d_i^< < h$ for some $h \leq a_i$ and that all cuts in feature $d_i^>$ have the from $d_i^> > h$ for some $h \geq a_i$. Thus, $e$ is put into the right subtree of each of these cuts. Since $e$ uses the default threshold in feature $d^*$, $e$ is put into the right subtree of the cut $d^* < 2$ which is a red leaf.

Consequently, in all cases $e$ is correctly classified as red by $T$.

*Step 2.4:* Consider a red edge example $e = (v_i^x, v_j^z)$. By assumption, $S$ is a multicolored independent set. Hence, at least one of the two endpoints $v_i^x$ and $v_j^z$ is not contained in $S$. Without loss of generality, assume that $v_i^x \notin S$ and that $i < j$. The argumentation is analog to the red choice examples $c_i^x$ where $x \neq a_i$: Without loss of generality, assume that $x < a_i$. By Step 1, $e$ will end up in the cut $d_i^< < 1.5$ of $T$. Also, recall that the next cuts in $T$ are $d_i^< < 2, \ldots, d_i^< < a_i$ in that specific order. Consequently, $e$ goes to the left subtree of the cut $d_i^< < x + 1/2$. If, $x = 1$, then the root of this left subtree is the cut $p_i^< < 2$. Otherwise, if $x \geq 2$, then the root of this left subtree is the cut $q_i^< < h$ for some $h \leq 2p - 1$ such that $h$ is even. Observe that in both cases $e$ is put into the left subtree of the cut $p_i^< < 2$ or $q_i^< < h$ which is a red leaf. Thus, $e$ is correctly classified as red.

*Step 2.5:* (A) Consider a red filler example $e = r_j^{i*}$. The argumentation is very similar to the red choice examples. First, consider the case $j \neq a_i$ and assume without loss of generality that $j < a_i$ (the case $j > a_i$ follows by considering feature $d_i^>$ instead). By Step 1, $e$ will end up in the cut $d_i^< < 1.5$ of $T$. Also, recall that the next cuts in $T$ are $d_i^< < 2, \ldots, d_i^< < a_i$ in that specific order. Next, observe that the cut $d_i^< < (j+1)/2$ puts $e$ into its left subtree. If $j = 1$, then the root of this subtree is $p_i^< < 2$ and puts $e$ into its left subtree which is a red leaf. Otherwise, if $j \geq 2$, then the root of this subtree is $q_i^< < j + 1$. If $j$ is even, cut $q_i^< < j + 1$ puts $e$ into its right subtree which is a red leaf. Otherwise, if $j$ is odd, cut $q_i^< < j + 1$ puts $e$ into its left subtree which is a red leaf.

Second, consider the case $j = a_i$. Again, observe that in $T$ all cuts in feature $d_i^<$ have the form $d_i^< < h$ for some $h \leq a_i$ and that all cuts in feature $d_i^>$ have the from $d_i^> > h$ for some $h \geq a_i$. Thus, $e$ is put into the right subtree of each of these cuts. Since $e$ uses the default threshold in feature $d^*$, $e$ is put into the right subtree of the cut $d^* < 2$ which is a red leaf.

Hence, $e$ is correctly classified as red.

(B) The argumentation for a blue filler example $e = b_j^{i*}$ is very similar: If $j \neq a_i$, then the cuts $p_i^< < 2$ and $q_i^< < j + 1$ will put $e$ into its blue leaf. Otherwise, if $j = a_i$, then $e$ ends up at the cut $d^* < 2$ and then $e$ is put into the left subtree which is a blue leaf. Consequently, $e$ is correctly classified as blue.

*Step 2.6:* (A) Consider a red dummy example $e = r_j^{i<}$. First, assume that $a_i \geq 2$. Then the first cut of $T$ in feature $d_i^<$ is $d_i^< < 1.5$. Hence, $e$ is put into the left subtree of the cut $d_i^< < 1.5$. The root of this subtree is the cut $p_i^< < 2$. Consequently, $e$ is put into the left subtree of the cut $p_i^< < 2$ which is a red leaf. Second, consider the case that $a_i = 1$. Then, the cuts of $T$ in feature $d_i^<$ are $d_i^< < -2p + 3, d_i^< < -2p + 4, \ldots, d_i^< < -1, d_i^< < 1$ in that specific order. If $j = 1$, then $e$ is put into the left subtree of the cut $d_i^< < -2p + 3$ whose root is the cut $p_i^< < 2$. Otherwise, if $j \geq 2$, then $e$ is put into the left subtree of the cut $d_i^< < -2p + 2 + j$ whose root is the cut $q_i^< < j + 1$. In both cases $e$ is then put into the red leaf of this cut.

The argumentation for a red dummy example $r_j^{i>}$ follows analogously with the features $d_i^>, p_i^>$, and $q_i^>$ instead of $d_i^<, p_i^<$, and $q_i^<$.

(B) The argumentation for the blue dummy examples $e = b_j^{i<}$ and $e = b_j^{i>}$ follows analogously as for the red dummy examples. The difference is that the cuts $p_i^< < 2$ and $q_i^< < j + 1$ put $e$ into the blue leaf of this cut.

Consequently, all dummy examples are correctly classified by $T$.

Thus, all examples are correctly classified by $T$ and we deal with a yes-instance for THRESHOLD ADJUSTMENT.

($\Leftarrow$) Let $T$ be a solution of the THRESHOLD ADJUSTMENT instance, that is, to up to $k = (p - 1) \cdot 2\kappa$ cuts of $T'$ a threshold adjustment operation was applied such that the resulting tree $T$ has $t = 0$ errors.

*Outline:* Observe that in $T'$ the classification paths of all blue separation examples $b(i, x)$ is identical to the classification paths of the red enforcing example. First, from that observation we conclude that in $T$ one of the following two cases has to occur: (a) $T$ contains a cut $d_i^< < h$ for some $h$ such that $x \le h + 1$ or (b) $T$ contains a cut $d_i^> > h$ for some $h \le x$. Second, we show that if $T$ contains a cut $d_i^< < h$ for some $h \in \{1.5, 2, 2.5, \ldots, p\}$, then $T$ also needs to contain the cuts $d_i^< < x$ for all $x \in \{1.5, 2, 2.5, \ldots, h\}$. Analogously we show that, if $T$ contains the cut $d_i^> > h$ for some $h \in \{1, 1.5, 2, \ldots, p - 1/2\}$, then $T$ also needs to contain the cuts $d_i^> > x$ for all $x \in \{h, h + 1/2, h + 1, \ldots, p - 1/2\}$. Third, this structure of cuts in $d_i^<$ and $d_i^>$ tightens the budget $k$ and allows us to define a selected vertex of the color class $i$. Finally, we argue that all selected vertices form a multicolored independent set.

*Step 1:* As discussed above, the classification paths of all blue separation examples $b(i, x)$ is identical to the classification paths of the red enforcing example $e^*$ in $T'$. Recall that $e[d_i^<] = x + 1/2 = e[d_i^>]$ and that $e^*$ uses the default thresholds in each feature. Also observe that in $T'$ all cuts in feature $d_i^<$ have the form $d_i^< < h$ for some $h \le 1$ and all cuts in feature $d_i^>$ have the form $d_i^> > h$ for some $h \ge p$. Moreover, recall that $e$ and $e^*$ are put into the right subtree of each cut in $d_i^<$ and $d_i^>$ in $T'$. Since $T$ makes no errors, we conclude that $T$ needs to have (a) a cut $d_i^< < h$ for some $h$ such that $x \le h + 1$ or (b) a cut $d_i^> > h$ for some $h \le x$, in order to put $e$ into the left subtree of this cut and consequently separate $e$ from $e^*$.

*Step 2:* Consider the set $Z$ of cuts $d_i^< < h$ in $T$ where $h \in \{1.5, 2, 2.5, \ldots, p\}$. Let $d_i^< < x$ be a cut in $Z$. We now argue (a) that $T$ also needs to contain all cuts $d_i^< < h$ for any $1.5 \le h < z$ and (b) that the cuts of $Z$ need to appear in $T$ in ascending order with thresholds, that is, first cut $d_i^< < 1.5$, then cut $d_i^< < 2$, and so on.

Assume towards a contradiction that (a) is wrong, that is, $T$ contains a cut $d_i^< < x$ but not the cut $d_i^< < x - 1/2$. Without loss of generality we assume that $x$ is an integer. Now observe that each cut in feature $d_i^<$ handles the following examples incidentally: (a) the red filler examples $r_{2x-5}^{i*}, r_{2x-4}^{i*}$ and (b) the blue filler examples $b_{2x-5}^{i*}, b_{2x-4}^{i*}$. That means that each cut of $T$ either puts all 4 of these examples into its left subtree or into its right subtree. This is true since by our assumption $T$ does not contain the cut $d_i^< < x - 1/2$. Moreover, observe that since $T$ contains the cut $d_i^< < x$, at some cut $d_i^< < h$ for $h \ge x$ all these 4 examples are put into the left subtree of the cut $d_i^< < h$. Now, observe that this left subtree consists of a single inner node with a cut in feature $p_i^<$ or $q_i^<$. Next, note that each cut in feature $p_i^<$ or $q_i^<$ can only separate exactly one pair $r_j^{i*}, b_j^{i*}$ of filler examples, but not two. Thus, at least one filler example gets misclassified by $T$, a contradiction to the fact that $t = 0$.

Thus, if $T$ contains the cut $d_i^< < x$ for $x \ge 2$, then $T$ also needs to contain the cut $d_i^< < x - 1/2$. Note that the above argumentation implies that the cut $d_i^< < x - 1/2$ needs to appear before the cut $d_i^< < x$ because otherwise still at least one filler example gets misclassified.

Analogously, one can show that if $T$ contains the cut $d_i^> > x$ for some $x \le p - 1$, then $T$ also needs to contain the cut $d_i^> > x + 1/2$ and that in $T$ cut $d_i^> > x + 1/2$ needs to appear before the cut $d_i^> > x$.

*Step 3:* Let $b(i, x)$ be the blue separation example such that $T$ contains a cut $d_i^< < x + 1$ and such that $x$ is maximal. Next, we distinguish the value of $x$. If $x = p - 1$, then since $T$ contains the cut $d_i^< < p$, by Step 2, $T$ also needs to contain the cuts $d_i^< < h$ where $1.5 \le h \le p$. Observe that these are $2(p - 1)$ cuts in total.

Otherwise, if $x \le p - 2$ (this includes the case that $x$ does not exist), then by Step 2, $T$ also needs to contain the cuts $d_i^< < h$ where $1.5 \le h \le x + 1$. Observe that these are $2x$ cuts in total.

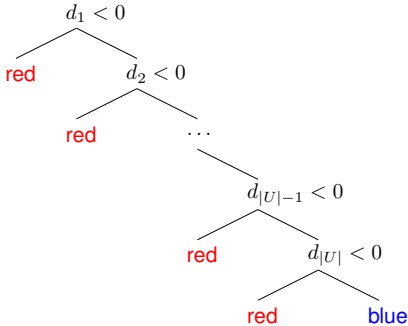

Figure 6: The initial decision tree in the reduction of Theorem 5.3 from HITTING SET.

Since $x \leq p - 2$, the blue separation example $b(i, x+1)$ exists. By Steps 1 and 2, $T$ contains (a) a cut $d_i^< < x + 1$ or (b) a cut $d_i^> > x$. By our assumption, (a) is not possible and thus $T$ contains the cut $d_i^> > x$. Again, by Step 2, we can conclude that $T$ also needs to contain the cuts $d_i^> > h$ for each $x \leq h \leq p - 1/2$. Note that these are $2(p - 1 - x)$ cuts in total. Thus, in total $T$ needs to perform at least $2x + 2(p - 1 - x) = 2(p - 1)$ threshold adjustment operations in features $d_i^<$ and $d_i^>$.

Recall that the budget is $k = 2(p-1)\kappa$. Since there are $\kappa$ color classes and in each pair $d_i^<$ and $d_i^>$ of features $T$ needs to perform at least $2(p-1)$ threshold adjustment operations, we conclude that to obtain $T$ exactly $2(p-1)$ threshold adjustment operations per features $d_i^<$ and $d_i^>$ are performed and that in no feature $q_i^<$ or $q_i^>$ any threshold adjustment operations is done.

If $x$ exists, we say that $v_i^{x+1} \in V_i$ is *selected*, and otherwise, if $x$ does not exist, we say that $v_i^1 \in V_1$ is *selected*. Let $S$ be the set of all selected vertices.

*Step 4:* We now argue that $S$ is a multicolored independent set. Clearly, $S$ contains exactly one vertex per color class. Hence, it remains to verify that $S$ is an independent set.

Observe that since $T$ has no classification errors, it is sufficient to show that $T$ cannot distinguish the blue forcing example $b^*$ and an red edge example $e = (v_i^x, v_j^z)$ if both endpoints $v_i^x$ and $v_j^z$ are contained in $S$: Let $e = e(v_i^x, v_j^z)$ be an edge example where $v_i^x, v_j^z \in S$. Observe that according to Steps 1-3 all cuts of $T$ in feature $d_i^<$ have the form $d_i^< < h$ for some $h \leq x$ and all cuts in feature $d_i^>$ have the from $d_i^> > h$ for some $h \geq x$. Hence, $e$ is put into the right subtree of each of these cuts. Analogously, we can argue that $e$ ends up in the right subtree of each cut in features $d_j^<$ and $d_j^>$. Since the blue forcing example $b^*$ uses the default thresholds in these 4 features, this is also true for $b^*$. Thus, both $e$ and $b^*$ end up in the cut $d^* < 2$. Both examples, however, are put in the left subtree of this cut. Thus, $T$ cannot distinguish the blue forcing example $b^*$ and the red edge example $e = (v_i^x, v_j^z)$. Consequently, $S$ is also an independent set.

**Lower Bound.** Recall that $d = 6 \cdot \kappa + 1, \delta_{\max} = 20$, and $t = 0$. Since MULTICOLORED INDEPENDENT SET is W[1]-hard with respect to $\kappa$ [3], we obtain that CUT EXCHANGE is W[1]-hard with respect to $d$ even if $\delta_{\max} = 20$, and $t = 0$. Furthermore, since MULTICOLORED INDEPENDENT SET cannot be solved in $f(\kappa) \cdot n^{o(\kappa)}$ time unless the ETH fails [3], we observe that CUT EXCHANGE cannot be solved in $f(d) \cdot |\mathcal{I}|^{o(d)}$ time if the ETH is true, where $|\mathcal{I}|$ is the overall instance size, even if $\delta_{\max} = 20$, and $t = 0$. □

### C.3 Proof of Theorem 5.3

*Proof.* We reduce from the W[2]-hard problem HITTING SET. Let $(U, \mathcal{S}, \kappa)$ be an instance of HITTING SET with a universe $U$ and a family of sets $\mathcal{S}$ such that we look for a hitting set of size $\kappa$. Essentially, we create a cut and a feature for each element of the universe and an example for each set of the instance.

Without loss of generality, assume that the elements of the universe are $1, 2, \ldots, |U|$. We create a THRESHOLD ADJUSTMENT instance with $|U|$ features $d_1, d_2, \ldots, d_{|U|}$, and for each set $S$ in $\mathcal{S}$ we create a red example $e$ such that $e[d_i] = 0$ if $i \in S$, and $e[d_i] = 1$ otherwise. We also create one

blue example $e'$ with $e'[d_i] = 1$ for all $i \in U$. Our initial decision tree has $|U|$ cuts $c_1, c_2, \ldots, c_{|U|}$ that form a chain where the right child of the cut $c_i$ is $c_{i+1}$ for all $i \in \{1, 2, \ldots, |U| - 1\}$. The right child of $c_{|U|}$ is a blue leaf. The left child of each cut is a red leaf. The feature of the cut $c_i$ is $d_i$ and the initial threshold is 0. An illustration of the initial decision tree is provided in Figure 6. Finally, let $k = \kappa$ and $t = 0$.

We claim that this instance of THRESHOLD ADJUSTMENT has a solution if and only if the HITTING SET instance has a solution. First, assume that the HITTING SET instance has a hitting set $H$. Then, adjusting the thresholds of cuts $c_i$ to 1 for all $i \in H$ results in 0 misclassifications: For the blue example, all thresholds are at most 1 and so it still goes to the blue leaf. Similarly for any red example $e$ corresponding to some set $S \in \mathcal{S}$, there is some cut $c_i$ with $i \in H$ whose threshold was adjusted to 1, and the example thus goes to a red leaf since $e[d_i] = 0$ by construction.

Assume now instead that the HITTING SET instance has no solution. Let us prove by contradiction that then the THRESHOLD ADJUSTMENT instance has no solution. Suppose the opposite, that is, there exists a solution by somehow adjusting some set of cuts $C$. First note that we cannot have increased any threshold above 1, since that would result in us misclassifying the only blue example if the leaf is not relabeled, and otherwise some red example gets misclassified. Consequently, the blue example ends up in the only leaf that is originally labeled as blue, and thus its label cannot be changed.

Consider now the set $H = \{i \colon c_i \in C\}$. It cannot be a hitting set, since our HITTING SET instance has no solution. Thus, there is some set $S \in \mathcal{S}$ with $S \cap H = \emptyset$. Take now the red example $e$ corresponding to $S$. The only was we can classify $e$ as red is that the threshold of $c_i$ is greater than $e[d_i]$. This is not possible for any $i \in S$, since those cuts remain unadjusted by the fact that $S \cap H = \emptyset$. For all $i \notin S$, we have that $e[d_i] = 1$, and no threshold could have been increased above 1 without misclassifying the blue example. Thus, $e$ has to get misclassified. By contradiction, the THRESHOLD ADJUSTMENT instance has no solution.

Note that the instance is not yet reasonable, since some leaves are initially empty. The following modification to the instance suffices for making the it reasonable here and for the following propositions of the section. Let $n'$ be the number of examples in the instance before making this adjustments. For each feature $d_i$ with $i \in [|U|]$, we create $n' + 1$ red examples $e$ with $e[d_j] = -\infty$ for $j \geq i$ and $e[d_j] = \infty$ otherwise. We create also $n' + 1$ blue examples $e$ with value $\infty$ for all features. Now, no leaf is empty, and each leaf has more than $n'$ new examples, so their labels match the majority. The proof of W[2]-hardness even with these new examples goes identically to as above.

We have shown that the above reduction is correct. Note that $k = \kappa$, $D = 2$ and $t = 0$ for the constructed instance. Thus, THRESHOLD ADJUSTMENT is W[2]-hard with respect to $k$ even if $D = 2$ and $t = 0$. It is known that HITTING SET cannot be solved in $\mathcal{O}((|U| + \sum_{S \in \mathcal{S}} |S|)^{o(\kappa)})$ time assuming ETH, and not in $\mathcal{O}(|U|^{\kappa - \epsilon})$ time for any $\epsilon > 0$ assuming SETH, see Corollary 14.23 and Theorem 14.42 by Cygan et al. [3] (note that DOMINATING SET straightforwardly reduces to HITTING SET). The ETH-based lower bound for THRESHOLD ADJUSTMENT follows because $k = \kappa$ and the constructed instance of THRESHOLD ADJUSTMENT has size polynomial in $|U| + |\mathcal{S}|$. Further, since $s = |U|$ the SETH-based lower bound for THRESHOLD ADJUSTMENT also follows. $\qquad\square$

In the following propositions of the section, we do not construct our reduced instances to be reasonable for conciseness. However, an adjustment identical to that of Theorem 5.3 suffices for making them reasonable up to and including Corollary 5.7.

### C.4 Proof of Corollary 5.4

*Proof.* The reduction is similar to Theorem 5.3, but we reduce from the NP-hard problem VERTEX COVER instead of HITTING SET. Take an arbitrary VERTEX COVER instance $(G, \kappa)$ such that we look for a vertex cover of size $\kappa$ from the graph $G = (V, E)$. We then perform the same reduction as in Thm. 5.3, treating $V$ as the universe and $E$ as the family of sets, each of which is of size two. $\quad\square$

### C.5 Proof of Corollary 5.5

*Proof.* The reduction is similar to Theorem 5.3 and Corollary 5.4 but we reduce from the W[1]-hard problem PARTIAL VERTEX COVER. Take an arbitrary PARTIAL VERTEX COVER instance $(G, \kappa, \tau)$

where we look for a subset of $\kappa$ vertices from the graph $G = (V, E)$ such that at most $\tau$ edges are uncovered. We perform the reduction similarly to Theorem 5.3, treating $V$ as the universe and $E$ as the family of sets, each of which is of size two. The differences to that reduction are as follows: let $t = \tau$ and create $t + 1$ blue examples $e$ with $e[d_i] = 1$ for all $i \in U$. This prevents us from increasing any threshold above 1 as we would get more than $t$ errors. On the other hand, we can now misclassify at most $\tau = t$ examples corresponding to the edges of $G$. The rest of the proof then goes analogously. $\qquad\square$

## C.6 Proof of Theorem 5.6

*Proof.* We again reduce from HITTING SET. This time, the idea is that each cut is initially in an *undecided* state (threshold 0) and has to be *included* (threshold 1) or *not included* (threshold $-1$) in the hitting set. Our initial decision tree is constructed the same way as in Theorem 5.3. However, our examples are different.

We add a single blue *universe example* $e$ for each element $u \in U$ such that $e[d_u] = -1$ and otherwise $e[d_i] = 1$ for all $i \in U$ with $i \neq u$. These examples are initially misclassified. For each set $S$ in $\mathcal{S}$, we create $t + 1$ copies of a red example $e$ such that $e[d_i] = 0$ if $i \in S$, and $e[d_i] = 1$ otherwise. These are also initially misclassified. Finally, we create $t + 1$ copies of a blue example $e'$ with $e'[d_i] = 1$ for all $i \in U$. These are initially correctly classified but get misclassified if any threshold is adjusted to be more than 1. Finally, set $k = |U|$ and $t = \kappa$.

We again claim that this THRESHOLD ADJUSTMENT instance has a solution if and only if the initial HITTING SET instance has a solution. The other direction goes almost identically to the proof of Theorem 5.3; if a suitable hitting set $H$ exists, we increase the threshold of the cut $c_i$ to 1 for all $i \in H$ and decrease the threshold to $-1$ for cuts $c_i$ with $i \in U \setminus H$. Consequently, we get $|H| \leq \kappa = t$ errors.

Assume now instead that the HITTING SET instance does not have a solution and suppose that THRESHOLD ADJUSTMENT has a solution. We know that at most $t = \kappa$ cuts can have threshold above $-1$, since otherwise more than $t$ universe examples get misclassified. We can argue identically to Theorem 5.3 that if $C$ is the set of cuts whose scores are above $-1$ and $H = \{i \colon c_i \in C\}$, then there exists some set $S \in \mathcal{S}$ that $H$ does not hit and the corresponding example $e$ is misclassified. Since there are $t + 1$ copies of that example, this would result in us exceeding the error budget. Thus, no solution to THRESHOLD ADJUSTMENT exists.

Thus the reduction is correct. The ETH and SETH-based lower bounds for THRESHOLD ADJUST-MENT follow in an analogous way to Theorem 5.3. $\qquad\square$

## C.7 Proof of Corollary 5.7

*Proof.* Direct adaptation of Theorem 5.6 by reducing from VERTEX COVER analogously to Corollary 5.4. $\qquad\square$

## D  Additional Material for Section 6

**Data sets and initial decision trees.** We used 47 datasets from the Penn Machine Learning Benchmarks library [38]. 32 of the datasets were used before for computing minimum-size trees [1, 32, 42] and since the number of examples was usually small we added further larger datasets. The datasets range from 72 to 5404 examples (mean 655.09, median 302.00); for the full details, see Table 2. We transformed the data sets as follows (similarly to Janota and Morgado [19]): First, we replaced each categorical feature by a set of new binary features indicating whether an example is in the category. Second, we converted each instance into a binary classification problem by making two classes, one of which contains all examples of the largest original class and one which contains all remaining examples. Finally, if two examples of different classes had the same value in all features, we removed one of them arbitrarily.

We computed unpruned and pruned trees using the C4.5 heuristic for decision-tree computation [36] implemented WEKA 3.8.5 [9]. The unpruned trees were obtained by running WEKA's J48 classi-fier with the flags `-no-cv -B -M 0 -J -U`. The pruned trees were computed by the *replacement heuristic* implemented in J48 when run with the flags `-no-cv -B -M 0 -J -S`. Overall, the tree

Table 2: Dataset statistics.

| Dataset | # Examples | # Features | Class 0 | Class 1 | Class Ratio |
|---|---|---|---|---|---|
| analcatdata_boxing2 | 132 | 3 | 61 | 71 | 1.16 |
| appendicitis | 106 | 7 | 85 | 21 | 0.25 |
| australian | 690 | 18 | 383 | 307 | 0.8 |
| backache | 180 | 55 | 155 | 25 | 0.16 |
| banana | 5300 | 2 | 2924 | 2376 | 0.81 |
| biomed | 209 | 14 | 75 | 134 | 1.79 |
| breast-cancer | 266 | 31 | 188 | 78 | 0.41 |
| bupa | 341 | 5 | 168 | 173 | 1.03 |
| cars | 392 | 12 | 147 | 245 | 1.67 |
| cleve | 302 | 27 | 164 | 138 | 0.84 |
| cleveland | 303 | 27 | 139 | 164 | 1.18 |
| cleveland-nominal | 130 | 17 | 61 | 69 | 1.13 |
| colic | 357 | 75 | 134 | 223 | 1.66 |
| contraceptive | 1358 | 21 | 764 | 594 | 0.78 |
| corral | 160 | 6 | 90 | 70 | 0.78 |
| dermatology | 366 | 129 | 254 | 112 | 0.44 |
| diabetes | 768 | 8 | 268 | 500 | 1.87 |
| ecoli | 327 | 7 | 184 | 143 | 0.78 |
| flare | 1066 | 10 | 884 | 182 | 0.21 |
| glass | 204 | 9 | 128 | 76 | 0.59 |
| glass2 | 162 | 9 | 86 | 76 | 0.88 |
| haberman | 283 | 3 | 73 | 210 | 2.88 |
| hayes-roth | 84 | 15 | 59 | 25 | 0.42 |
| heart-c | 302 | 27 | 138 | 164 | 1.19 |
| heart-h | 293 | 29 | 106 | 187 | 1.76 |
| heart-statlog | 270 | 25 | 150 | 120 | 0.8 |
| hepatitis | 155 | 39 | 123 | 32 | 0.26 |
| Hill_Valley_without_noise | 1212 | 100 | 600 | 612 | 1.02 |
| hungarian | 293 | 29 | 187 | 106 | 0.57 |
| ionosphere | 351 | 34 | 126 | 225 | 1.79 |
| irish | 500 | 5 | 278 | 222 | 0.8 |
| lupus | 86 | 3 | 52 | 34 | 0.65 |
| lymphography | 148 | 50 | 67 | 81 | 1.21 |
| molecular_biology_promoters | 106 | 228 | 53 | 53 | 1.0 |
| monk2 | 601 | 6 | 395 | 206 | 0.52 |
| new-thyroid | 215 | 5 | 65 | 150 | 2.31 |
| phoneme | 5404 | 5 | 3818 | 1586 | 0.42 |
| pima | 768 | 8 | 500 | 268 | 0.54 |
| postoperative-patient-data | 72 | 22 | 50 | 22 | 0.44 |
| prnn_synth | 250 | 2 | 125 | 125 | 1.0 |
| profb | 672 | 9 | 448 | 224 | 0.5 |
| schizo | 340 | 14 | 140 | 200 | 1.43 |
| soybean | 622 | 133 | 545 | 77 | 0.14 |
| tae | 106 | 5 | 71 | 35 | 0.49 |
| titanic | 2099 | 8 | 1418 | 681 | 0.48 |
| tokyo1 | 959 | 44 | 346 | 613 | 1.77 |
| yeast | 1479 | 8 | 1050 | 429 | 0.41 |

size $s$ ranges from 2 to 607 (mean 71.23, median 29.00); the number of features $d$ ranges from 1 to 88 (mean 11.35, median 8.00); the domain size $D$ ranges from 1 to 321 (mean 17.26, median 6.00); the number of classification errors ranges from 0 to 302 (mean 20.66, median 1.00). For the full details, see Table 3.

Table 3: Decision trees used in our experiments: The first entry is for the unpruned tree $T$, the second for the tree $T$ computed by the replacement heuristic.

| Dataset | Size $s$ | # Features $d$ used in $T$ | Domain $D$ used in $T$ | Errors |
|---|---|---|---|---|
| analcatdata_boxing2 | 48 / 9 | 3 / 3 | 11 / 5 | 0 / 17 |
| appendicitis | 15 / 10 | 6 / 6 | 5 / 3 | 0 / 2 |
| australian | 90 / 46 | 13 / 11 | 29 / 11 | 0 / 22 |
| backache | 26 / 13 | 13 / 9 | 7 / 2 | 0 / 7 |
| banana | 607 / 186 | 2 / 2 | 321 / 107 | 1 / 249 |
| biomed | 21 / 3 | 6 / 3 | 6 / 1 | 0 / 15 |
| breast-cancer | 95 / 31 | 25 / 21 | 8 / 6 | 2 / 31 |
| bupa | 111 / 72 | 5 / 5 | 21 / 18 | 0 / 25 |
| cars | 22 / 15 | 7 / 7 | 7 / 5 | 0 / 3 |
| cleve | 57 / 29 | 15 / 13 | 12 / 5 | 0 / 15 |
| cleveland | 55 / 31 | 16 / 13 | 10 / 6 | 0 / 13 |
| cleveland-nominal | 46 / 8 | 15 / 8 | 1 / 1 | 6 / 23 |
| colic | 51 / 28 | 27 / 18 | 6 / 4 | 0 / 15 |
| contraceptive | 486 / 120 | 21 / 21 | 33 / 21 | 10 / 217 |
| corral | 13 / 13 | 5 / 5 | 1 / 1 | 0 / 0 |
| dermatology | 5 / 3 | 4 / 3 | 1 / 1 | 0 / 2 |
| diabetes | 137 / 96 | 8 / 8 | 24 / 16 | 0 / 24 |
| ecoli | 25 / 5 | 5 / 3 | 11 / 2 | 0 / 10 |
| flare | 93 / 15 | 8 / 7 | 5 / 4 | 125 / 159 |
| glass | 28 / 26 | 7 / 7 | 7 / 7 | 0 / 1 |
| glass2 | 22 / 16 | 6 / 5 | 6 / 5 | 0 / 4 |
| haberman | 92 / 21 | 3 / 3 | 30 / 9 | 2 / 38 |
| hayes-roth | 14 / 12 | 11 / 10 | 1 / 1 | 0 / 1 |
| heart-c | 57 / 29 | 15 / 13 | 12 / 5 | 0 / 15 |
| heart-h | 57 / 32 | 20 / 18 | 13 / 6 | 0 / 14 |
| heart-statlog | 54 / 27 | 17 / 13 | 13 / 7 | 0 / 15 |
| hepatitis | 18 / 12 | 10 / 9 | 3 / 2 | 0 / 3 |
| Hill_Valley_without_noise | 250 / 228 | 88 / 85 | 63 / 47 | 0 / 11 |
| hungarian | 57 / 32 | 19 / 19 | 13 / 6 | 0 / 14 |
| ionosphere | 21 / 19 | 12 / 11 | 4 / 4 | 0 / 1 |
| irish | 2 / 2 | 1 / 1 | 2 / 2 | 0 / 0 |
| lupus | 25 / 4 | 2 / 2 | 20 / 2 | 0 / 13 |
| lymphography | 23 / 14 | 18 / 11 | 1 / 1 | 0 / 5 |
| molecular_biology_promoters | 12 / 10 | 11 / 9 | 1 / 1 | 0 / 1 |
| monk2 | 40 / 40 | 6 / 6 | 1 / 1 | 0 / 0 |
| new-thyroid | 13 / 9 | 5 / 5 | 4 / 4 | 0 / 2 |
| phoneme | 504 / 341 | 5 / 5 | 159 / 99 | 0 / 98 |
| pima | 137 / 96 | 8 / 8 | 24 / 16 | 0 / 24 |
| postoperative-patient-data | 23 / 21 | 13 / 13 | 1 / 1 | 0 / 1 |
| prnn_synth | 39 / 9 | 2 / 2 | 24 / 5 | 0 / 23 |
| profb | 185 / 115 | 9 / 9 | 24 / 17 | 0 / 42 |
| schizo | 83 / 69 | 12 / 12 | 15 / 10 | 0 / 8 |
| soybean | 28 / 13 | 22 / 12 | 1 / 1 | 0 / 8 |
| tae | 41 / 21 | 5 / 5 | 13 / 7 | 0 / 11 |
| titanic | 336 / 61 | 8 / 8 | 61 / 20 | 157 / 302 |
| tokyo1 | 46 / 34 | 24 / 22 | 10 / 5 | 0 / 6 |
| yeast | 315 / 125 | 8 / 7 | 43 / 24 | 0 / 129 |

**Further pruning pruned trees.** The above indicates that misclassifications may not be strongly reducable by local search. We next wanted to know whether it is possible to further prune the trees after local search, keeping the number of misclassifications at most the same. Indeed, in the solved problems, for 18 it was possible to prune further after at most two adjustments and for 24 this was possible after two exchanges. The average nodes pruned were 1.40 and 1.78, respectively. In one case, the tree size could be almost halved after one exchange. The full details for the improvable instances are shown in Table 4.

**Combining pruning with local search.** Next, we wanted to know whether we improve pruning perfomance over the heuristics. Thus, we started with the unpruned heuristically computed trees,

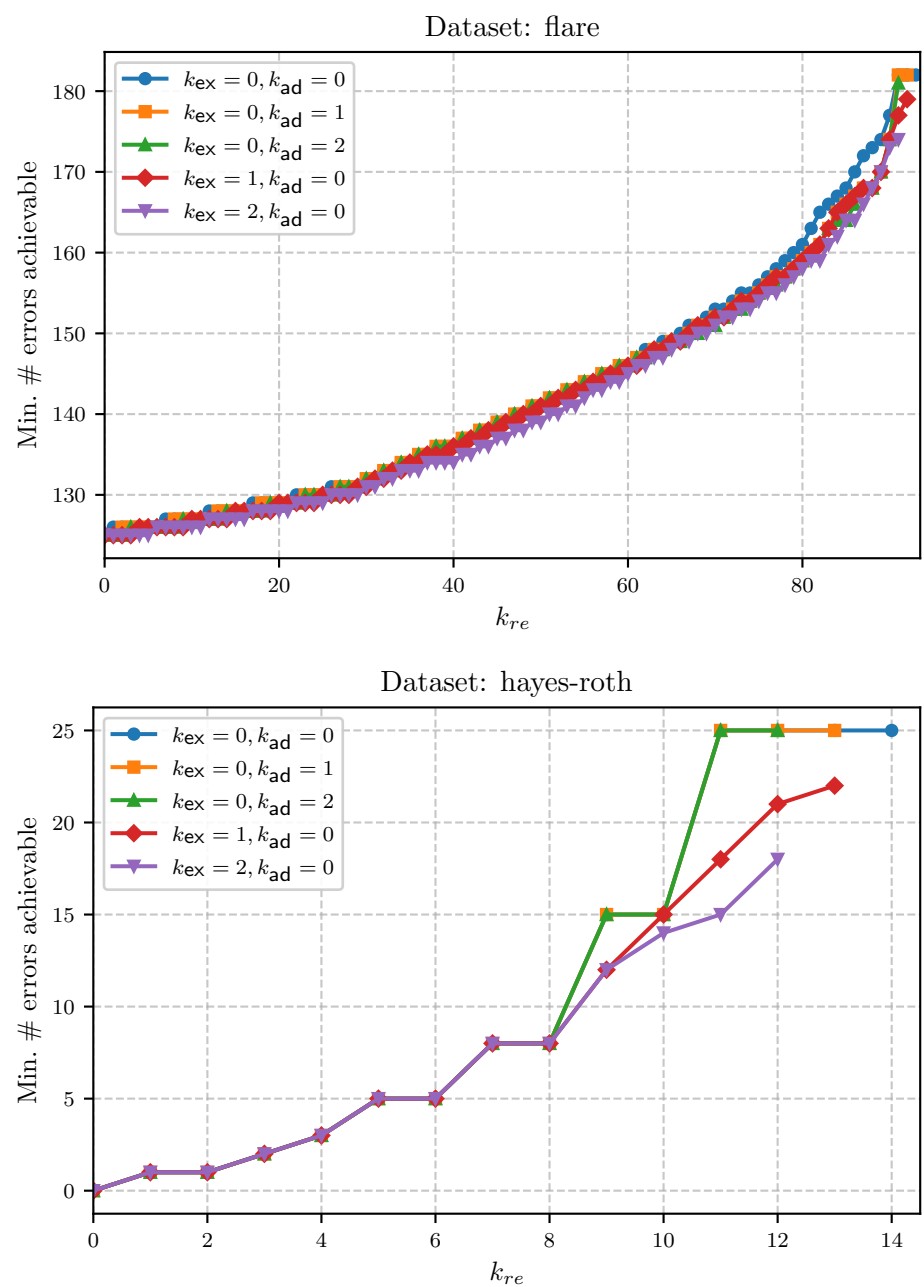

Figure 7: Pareto fronts for number of achievable errors vs. number of pruning operations $k_{\text{re}}$ for various local search parameters.

Table 4: Number of nodes for decision trees that can be pruned without increasing errors after local search. Dashes indicate timeouts.

| Dataset | Tree Size | Initial Errors | $k_{\mathsf{ad}} = 1$ | $k_{\mathsf{ad}} = 2$ | $k_{\mathsf{ex}} = 1$ | $k_{\mathsf{ex}} = 2$ |
|---|---|---|---|---|---|---|
| banana | 607 | 249 | 1 | – | – | – |
| breast-cancer | 95 | 31 | 1 | 1 | 1 | 2 |
| bupa | 111 | 25 | 1 | – | 1 | – |
| cleve | 57 | 15 | 2 | 2 | 2 | – |
| cleveland | 55 | 13 | 2 | 2 | 2 | – |
| cleveland-nominal | 46 | 23 | 0 | 0 | 1 | 1 |
| colic | 51 | 15 | 0 | 0 | 1 | – |
| contraceptive | 486 | 217 | 1 | 1 | 1 | – |
| corral | 13 | 0 | 0 | 0 | 6 | 6 |
| diabetes | 137 | 24 | 1 | – | 1 | – |
| ecoli | 25 | 10 | 1 | 2 | 1 | 2 |
| flare | 93 | 159 | 0 | 0 | 0 | 1 |
| glass | 28 | 1 | 1 | – | 1 | – |
| glass2 | 22 | 4 | 0 | 1 | 0 | – |
| haberman | 92 | 38 | 1 | 1 | 1 | 1 |
| heart-c | 57 | 15 | 2 | 2 | 2 | – |
| heart-h | 57 | 14 | 0 | 0 | 1 | – |
| heart-statlog | 54 | 15 | 1 | 1 | 1 | – |
| hungarian | 57 | 14 | 0 | 0 | 1 | – |
| ionosphere | 21 | 1 | 1 | – | – | – |
| lymphography | 23 | 5 | 0 | 0 | 0 | 1 |
| pima | 137 | 24 | 1 | – | 1 | – |
| postoperative-patient-data | 23 | 1 | 0 | 0 | 0 | 1 |
| profb | 185 | 42 | 1 | – | 1 | – |
| soybean | 28 | 8 | 0 | 0 | 2 | – |
| tae | 41 | 11 | 0 | 1 | 0 | 1 |
| yeast | 315 | 129 | 1 | – | 2 | – |

pruned them to the same size as the heuristically computed pruned trees while also allowing up to two local search operations in the process. In the solved problems, this was the case for 10 trees with adjustments and 17 with exchanges, where the errors were improved by 5.16% and 8.92% on average, respectively. The full details are shown in Table 5. Since the heuristic chooses only one possible tradeoff between errors and number of pruned nodes, we also looked at the full pareto fronts for these goals, defined by zero local search operations, one or two. A plot of the pareto fronts for two representative instances is shown in Figure 7. The behaviour of error decreases for local search is relatively consistent over different tradeoffs but local search becomes much more effective for large numbers of pruned nodes, especially for cut exchange.

**Summary.** Overall, we have indication that our algorithms can feasibly be used to gain new insights into the amenability of small real-world decision trees. It seems that the heuristics choose tradeoffs between pruning and errors that are only very weakly amenable to local search. (Note that such observations are only possible to make with algorithms that find the optimum values, as we have designed here for the first time.) When pruning more nodes than the heuristics commonly choose, local search may be effective and important.

Table 5: Error numbers obtainable when combining pruning with local search. Dashes indicate timeouts.

| Dataset | Pruned Nodes | Heur. Errors | $k_{ad} = 1$ | $k_{ad} = 2$ | $k_{ex} = 1$ | $k_{ex} = 2$ |
|---|---|---|---|---|---|---|
| australian | 44 | 22 | 22 | – | 21 | – |
| breast-cancer | 64 | 31 | 30 | 30 | 30 | – |
| cleve | 28 | 15 | 14 | 14 | 14 | – |
| cleveland | 24 | 13 | 12 | 12 | 12 | – |
| cleveland-nominal | 38 | 23 | 23 | 23 | 22 | 22 |
| colic | 23 | 15 | 15 | 15 | 14 | – |
| contraceptive | 366 | 217 | 217 | – | 216 | – |
| dermatology | 2 | 2 | 2 | 2 | 2 | 1 |
| ecoli | 20 | 10 | 8 | 8 | 8 | – |
| flare | 78 | 159 | 157 | 157 | 157 | 156 |
| haberman | 71 | 38 | 37 | 37 | 37 | 37 |
| heart-c | 28 | 15 | 14 | 14 | 14 | – |
| lymphography | 9 | 5 | 5 | 5 | 5 | 4 |
| profb | 70 | 42 | 41 | – | 41 | – |
| soybean | 15 | 8 | 8 | 8 | 7 | – |
| titanic | 275 | 302 | 301 | – | 301 | – |
| yeast | 190 | 129 | 127 | – | 126 | – |

