# OpenReview forum: "Improving Decision Trees through the Lens of Parameterized Local Search"
_NeurIPS.cc/2025/Conference — NeurIPS 2025 poster_

### Official Review · Reviewer_btpa · 2025-06-21

**Clarity:** 3
**Significance:** 2
**Originality:** 2
**Rating:** 3
**Confidence:** 3

**Summary:**

This paper investigates the problem of improving decision trees through parameterized local search, focusing on two canonical operations: adjusting the threshold of an internal node and swapping its feature with a new one. While both problems are shown to be NP-complete in general, the paper provides a detailed analysis of their parameterized complexity across a wide range of structural and data-dependent parameters. The authors identify several tractable parameter combinations, most notably proving that the problem becomes fixed-parameter tractable when the number of features and the maximum domain size are both small. Efficient algorithms are proposed for these cases, including a dynamic programming method and a combined approach that incorporates pruning. A proof-of-concept implementation is evaluated on 47 benchmark datasets, showing that standard greedy decision tree learners already yield models that are nearly optimal in their local neighborhoods, with only marginal gains observed through local search.

**Questions:**

1. Can you elaborate on the intuition behind why improving a tree locally can be harder than learning one from scratch in some parameter regimes?

2. Given that local search leads to only minor improvements in most experiments, do you view this as a sign of strong heuristics or limitations of the local neighborhood?

3. Would your framework extend to other local operations such as subtree swapping or merging? Are any such cases tractable?

4. In practice, what ranges of d and D do you consider manageable for your exact algorithms to run efficiently?

**Ethical Concerns:**

["NO or VERY MINOR ethics concerns only"]

**Final Justification:**

While I appreciate the clarifications and author engagement with the questions raised, several concerns remain either unaddressed or insufficiently acknowledged in the current framing of the paper. The empirical gains are minimal, and framing this as a strength seems overstated without clearer practical implications or guidance for future work. The restriction to only two local operations limits the generality of the results, and the omission of more structural moves weakens claims about the overall difficulty of local tree optimization. Scalability remains a concern, and the implementation, while a useful proof of concept, lacks clear potential for practical deployment. Lastly, accessibility remains limited due to dense parameterized complexity framing with little effort to connect to typical ML workflows. The contribution is solidly theoretical but should be more clearly positioned as such.

**Limitations:**

yes

**Quality:**

3

**Strengths And Weaknesses:**

**Strengths**:

- Provides a rigorous, systematic parameterized complexity map for two decision-tree improvement problems.

- Identifies non-trivial FPT results and proves W[1]/W[2]-hardness under other settings.

- Offers new algorithmic contributions, including exact dynamic programs and pruning-aware extensions.

- Connects and extends recent theoretical results in decision tree learning and pruning.

- Clarity of exposition is high for a theory-heavy paper; good use of figures and algorithm pseudocode.

**Weaknesses**:

- Empirical improvements from local search are very limited (1–2 errors on most datasets).

- Theoretical algorithms do not scale well; many instances time out for k=2.

- Focused only on two types of local moves; omits more structural modifications, e.g., subtree switches.

- Heavy reliance on parameterized complexity may limit accessibility to broader ML audiences.

---

> ### Author Rebuttal · Authors · 2025-07-31
>
> Thank you for your careful evaluation!
>
> Before addressing your questions, allow us a comment:
>
> You mention that the fact that empirical improvements from local
> search are very limited is a weakness. We consider this a strength:
> This is an important finding that is impossible without algorithms
> that are guaranteed to find the optimum in the local neighborhood and
> it informs future research and has direct implications for practice.
> I.e. it is likely not necessary and useful to locally optimize
> decision trees computed by standard heuristics except in the regimes
> outlined in our paper. Our resources are more usefully spent
> elsewhere.
>
> As to your questions:
>
> **1. intuition behind why improving a tree locally can be harder than
>    learning one from scratch:**
>
> - To clarify this, our reduction from learning to local search basically
> uses scaffolding to construct a useless decision tree which then has
> to be made useful via changing its cuts with local-search operations.
> In the reduction the number of local-search operations roughly
> corresponds to the size of the tree to be learned. Thus, on the one
> hand local-search in a k-step neighborhood is harder than learning a
> size-k tree but on the other hand in practice the number of local
> search operations will be much lower than the tree size to be learned.
>
> - Note, however, that this intuition is not universally true, since for
> instance for parameter s + t, Threshold Adjustment is fixed-parameter
> tractable whereas Decision Tree Learning is W[1]-hard for s even for t
> = 0 (see Ref. [31, Thm 2]).
>
> **2. local search leads to only minor improvements in most experiments,
>    do you view this as a sign of strong heuristics or limitations of
>    the local neighborhood?**
>
> - We think it is rather a strength of the considered heuristic: Since
> these local-search operations are applied in the many cited sources,
> there must be many situations where they do improve the trees
> considered there effectively. This can be for two reasons: The trees
> in these applications are worse to start with or their heuristics are
> able to search a subset of a much larger neighborhood which happens to
> contain better trees. Both implicit questions would make an
> interesting follow-up work.
>
> **3. other local operations such as subtree swapping or merging**
>
> - We think this is interesting but since this is the first study on this
> topic, we had to focus our ressources on the seemingly most-used
> operations (and still, note the length and content of our paper).
> Subtree merging and swapping seem quite different in their structure
> and thus many of our techniques will likely not be directly
> applicable. Some ideas from the dynamic program for parameters d and D
> might transfer over, however.
>
> **4. In practice, what ranges of d and D do you consider manageable?**
>
> - Note that we wouldn't consider our proof-of-concept implementation
> practical yet. There is huge potential for improvement, both on the
> implementation side (since it is in Python) and algorithmic side (we
> think there are many possible heuristic improvements to the code that
> maintain optimality which we did not have time for). That said, the
> current implementation usually completes within 1h if d_T + D_T <= 20
> (with outliers in both directions), where d_T is the largest number of
> dimensions on a root-leaf path in the input tree and D_T the largest
> number of thresholds on a root-leaf path. This implies it completes if
> d + D <= 20 but these can be even larger.
>
> **Further comments:**
>
> - **poor scalability:** We agree, but note that this is a proof of concept
>   that is supposed to show that the algorithm is implementable and
>   that it already yields results on standard benchmark data without
>   much engineering. Note that with efficiency improvements and larger
>   data we don't expect the running time to scale strongly with
>   increased k, as the theory shows. Still, our main contribution is
>   the theory.
>
> - **parameterized complexity may limit accessibility to broader ML
>   audiences:** Please note that this paper follows an ongoing vibrant
>   tradition of parameterized complexity analysis at top-level ML
>   venues, see References [7, 10, 12, 22, 23, 30, 31, 38] just for
>   decision trees and closely related problems.

---

> > ### Comment · Reviewer_btpa · 2025-08-02
> >
> > Thank you for the detailed rebuttal. Your clarifications are appreciated and help contextualize some of the decisions and findings in the paper.

---

### Official Review · Reviewer_GnjU · 2025-06-26

**Clarity:** 3
**Significance:** 2
**Originality:** 3
**Rating:** 3
**Confidence:** 3

**Summary:**

This paper studies optimizing decision trees by performing a fixed number of local-search operations.
They define two types of local-search operations:

1. Adjusting the threshold of a cut
2. Exchanging the feature of that cut (and setting a new threshold)

The authors show that these two subproblems are NP-complete in general.
They further investigate the parameterized complexity of these subproblems with respect to a bunch of parameters.
They also found that some combination of parameters can make the problems FPT.

However, I found that the studied subproblems are not really "local", and its complexity is reluctantly linked to the hardness of learning a global tree.

**Questions:**

W1 and W2:
The authors define the two subproblems as minimizing the classification errors after $k$-step local-search operations.
The $k$ steps relate the subproblems to the well-known learning problem for a full decision tree.
This looks like a somewhat abusive use of local operations.

Although $k$-step local search indeed exists in the literature, I believe it is not the most natural and popular one.
The first local-search operation of interest should be the 1-step operation.
It is also the most computationally feasible one.

How do we efficiently solve the 1-step subproblem?
Is it hard?
What is the effect of optimizing a tree by a sequence of 1-step operations?
The authors skip these questions and go directly to the $k$-step version.
This renders the study incomplete in my opinion.

**Ethical Concerns:**

["NO or VERY MINOR ethics concerns only"]

**Final Justification:**

Thank you for the rebuttal.
Thank you for pointing me to the experiments where "heuristically computed trees were 2-optimal or close to 2-optimal".
My concern about the non-local search has been addressed.
I suggest the authors motivate the k-step local search differently in the revised paper to avoid confusion.

I also agree with other reviewers that the paper in its current form has a small impact on the applicability.

I slightly improved my score.

**Limitations:**

yes

**Quality:**

2

**Strengths And Weaknesses:**

Strengths:

1. Theoretical analysis of the parameterized complexity of two local-search operations for learning decision trees.
2. The analysis is comprehensive by involving a bunch of parameters.
3. Giving dynamic programming algorithms for the two subproblems with respect to a combination of some parameters.

Weaknesses:

1. The definition of the two subproblems is not natural (See below).
2. A most straightforward 1-step local-search operation is not discussed in the paper.
3. FPT algorithms are mainly of theoretical interest, and are less relevant in practice.

---

> ### Author Rebuttal · Authors · 2025-07-31
>
> Thank you for your feedack!
> Below, we separately respond to your questions.
>
>
> **How do we efficiently solve the 1-step subproblem?**
>
> - The 1-step subproblem corresponds to the special case k=1. Consequently, according to Proposition 4.6, both Threshold Adustment and Cut Exchange can be solved in polynomial time.
> In our experimental study we investigated whether heuristically computed decision trees can be improved in terms of the number of errors by k=1 or k=2 modifications. Our results show that this is usually not the case. Consequently, allowing for a sequence of 1-step operations does usually not improve a decision tree since usually even a single 1-step modification yields no improvement.
>
>
> **Although k-step local search indeed exists in the literature**
>
> - We completly agree that the 1-step operation is the most important one. As we have written in our response to the previous question, considering 1-step operations is usually not sufficient to improve a tree computed by a heuristic. Hence, it is necessary to consider larger neighborhoods and to develop efficient algorithms for this case which we did in our work.
> Similar concepts of searching in a larger neighborhood are for example used in MCMC where the convergence of algorithms such as Metropolis–Hastings to the stationary distribution can be speed up by allowing more complex modifications to the current state of the Markov chain.
>
>
>
>
>
>
>
> **W1 and W2: The authors define the two subproblems as minimizing the classification errors after k-step local-search operations.**
>
> - Our operations are local in the sense of neighborhoods, not in the changes that are made in some local area of the tree. More precisely, consider a graph where each node is a decision tree and two vertices (decision trees) D_1 and D_2 are adjcant if tree D_2 can be produced by applying k threshold adjustments/cut exchanges to D_1. Now, the neighborhood of a vertex (decision tree) is the set of decision trees which can be produced with k local search operations. Increasing parameter k yields a larger neighborhood. Consequently, we think our neighborhood definition is quite natural.
>
>
> **FPT algorithms are mainly of theoretical interest, and are less relevant in practice.**
>
> - While the majority of works regarding FPT-algorithms only focus on theory, a lot of FPT-algorithms pave the way towards fast implementations, for example consider the problem of computing a minimal-size decision tree: First an FPT-algorithm with inpractical running time was presented (see reference 31 of our paper), which then triggered the development of another FPT-algorithm (see reference 23) which then was the foundation of the currently fasted implementation to compute minimal-size decision trees (see reference 39).
> Also, some of our algorithms, for example the d+D are of practical interest since without this algorithm we could not discover that the trees computed by common heuristics are locally optimal.

---

> > ### Comment · Reviewer_GnjU · 2025-08-03
> >
> > Thank you for the rebuttal. Thank you for pointing me to the experiments where "heuristically computed trees were 2-optimal or close to 2-optimal". My concern about the non-local search has been addressed. I suggest the authors motivate the k-step local search differently in the revised paper to avoid confusion.
> >
> > I also agree with other reviewers that the paper in its current form has a small impact on the applicability.
> >
> > I slightly improved my score.

---

> > > ### Author Response · Authors · 2025-08-07
> > >
> > > Thanks very much!
> > >
> > > Regarding applicability, note that we responded to these concerns to the other reviewers. For instance, Reviewer eGr1 was concerned that the conducted experiments suggest little applicability for the proposed local search problems. Here is part of our response:
> > > - We would like to mention that this conclusion was not possible before. Prior to our work there was no evidence whether the trees computed by these heuristics are locally optimal with respect to a small number of threshold adjustments and cut exchanges. Only thanks to our algorithmic study, we could develop efficient algorithms for this task. Moreover, only our implementations based on these algorithms allowed making this experiment. As shown by our complementing hardness results, this is quite surprising and in the future one should investigate this phenomenon further, that is, find explanations as to why the trees computed by these heuristics are often locally optimal for small values of k.
> > >
> > > For additional details such as concerns about scalability, refer to our responses to Reviewer eGr1 and btpa (opening comment of the response and paragraph “poor scalability”).

---

### Official Review · Reviewer_eGr1 · 2025-06-30

**Clarity:** 3
**Significance:** 1
**Originality:** 2
**Rating:** 4
**Confidence:** 3

**Summary:**

The paper provides a parameterized complexity analysis of local search post-processing for decision trees, specifically through cut exchange and threshold adjustment operations. Threshold adjustment allows the thresholds of internal decision tree nodes to be modified, whereas cut exchange additionally allows for node features to be exchanged.

Specific theoretical contributions include (1) an extension of previous W[1]-hardness result for decision tree construction under a binary feature domain, (2) introduction and formal connection of the cut exchange and threshold adjustment problems, (3) W[1], W[2], and NP hardness results for both problems across several cases, and (4) FPT and XP algorithms for both problems w.r.t various parameter combinations.

Beyond these contributions, a small-scale empirical investigation was also conducted. Results show that local modifications (<2 operations) were not able to significantly improve heuristically constructed trees, suggesting that such algorithms may not be broadly applicable and that heuristically constructed trees are optimal within limited local domains. The latter may help to explain challenges faced by other decision tree construction mechanisms based on local search (e.g., Markov chain Monte Carlo).

**Questions:**

Q1 In what settings might the proposed local search algorithms (or variants) prove empirically useful?

Q2 What broader connections could be made between this work and subfields such as general decision tree learning, greedy heuristics, post-processing, etc?

**Ethical Concerns:**

["NO or VERY MINOR ethics concerns only"]

**Final Justification:**

Thank you for the responses to myself and others. I am convinced that (1) the technical contributions made by this paper are more rigorous and significant than I previously expected and that (2) the hardness results for local optimization combined with the empirical results showing minimal potential upside of such optimization could have more far reaching implications to the field of decision tree learning. I have increased my score from 3 to 4 as a result.

**Limitations:**

yes

**Paper Formatting Concerns:**

The I used on lines 42 and 44 is not calligraphic like the I used to denote input in the abstract.

**Quality:**

3

**Strengths And Weaknesses:**

The paper is clear and well-formatted though somewhat verbose in its notation.

The significant theoretical contributions of the paper appear novel, breaking ground on the complexity of local search in decision tree construction. The strengthened W[1]-hardness results for general decision tree construction in the binary domain may also be of independent interest.

This said, the conducted experiments suggest little applicability for the proposed local search problems (cut exchange and threshold adjustment) for which most of the parameterized complexity analysis was conducted.

Given this limitation and a lack of stated impactful future extensions, I would not recommend this paper to be accepted in its current form. This said, I could be convinced provided evidence of impactful future extensions or further examples of connections or implications for general decision tree learning.

---

> ### Author Rebuttal · Authors · 2025-07-31
>
> Thank you for for your review and your feedback!
> We also want to thank the reviewer for pointing out this curious similarity between the issues of local search and MCMC.
> We respond to the Reviewer's concerns and questions separately below.
>
> **The conducted experiments suggest little applicability for the proposed local search problems**
>
> - We would like to mention that this conclusion was not possible before. Prior to our work there was no evidence whether the trees computed by these heuristics are locally optimal with respect to a small number of threshold adjustments and cut exchanges. Only thanks to our algorithmic study, we could develop efficient algorithms for this task. Moreover, only our implementations based on these algorithms allowed making this experiment. As shown by our complementing hardness results, this is quite surprising and in the future one should investigate this phenomenon further, that is, find explanations as to why the trees computed by these heuristics are often locally optimal for small values of k.
>
> - Also, in our experiments we showed that combining pruning and local search is better than doing this sequentially, that is, first prune the tree and then apply local search. This combined approach might be useful in setting where even a small improvement in the number of errors can be quite important, for example in healthcare.
> Moreover, threshold adjustment and cut exchange are used extensively in genetic algorithms (see references 18-21, 24-26, and 27 of our paper). Our algorithms could be quite helpful there to lead to much better rate of convergence. We did not test this hypothesis since it's beyond the scope of our paper. Moreover, the precise setting is not as clear as for the heuristics which we tested since many different genetic algorithms exist and its not clear which of them should be included in our experiments.
>
>
> **What broader connections could be made between this work and subfields such as general decision tree learning, greedy heuristics, post-processing, etc?**
>
> - We could imagine that the FPT-algorithm for parameter d+D which we used in our experiments can be extended to also cover other local search operations which modify the structure of the tree. We already showed (Theorem 4.3) that pruning and local search can be combinded in a single algorithm. As written above, our experimental results showed that this combined approach yields better results in terms of errors than applying both sequentially. This insight could help in developing new post-processing heuristics which lead to better results (same size but less errors) than the existing ones.
>
> - Also, since our new approach yields optimal solutions for local serach of decision trees, our algorithms could be used to develop new genetic algorithms with more powerful crossover operations.
> Moreover, currently minimal-size decision trees can only be computed for size up to s=20 (see citation 39 of our paper). A major hurdle to also solve instances where s>20 is the lack of good lower bounds to abort parts of the branch-and-bound approach early. Our local search algorithm could help in strengthening existing lower bounds and thus make the learning of minimal-size decision trees more applicable.

---

### Official Review · Reviewer_9RB4 · 2025-07-04

**Clarity:** 2
**Significance:** 3
**Originality:** 3
**Rating:** 5
**Confidence:** 4

**Summary:**

This work studies the problem of doing a fixed number of heuristic local-search algorithms like adjusting the threshold of an existing cut or changing the feature of a cut after a tree has been already learned. First, this work shows that these are NP-complete in general and hence, provides a fixed parameter parameterized complexity. For instance, they show that the problems remain hard for a small number d of features or small domain size D but the combination of both yields fixed-parameter tractability.

**Questions:**

- If we exchange feature of a cut and that feature appears as a sub child of that node, do we leave it unchanged?
- What about other local search operations like removing a leaf from the tree?
- I am confused about line 71 where the authors say that the local algorithms are optimal but the error decrease is mild. Could the authors please clarify that?
- From table 1, it seems that local search algorithms don’t help that much. Why are they so popular in practice if that is the case?
- Could the authors comment on the algorithms runtime that they have provided in terms of what they can hope to further improve?

**Ethical Concerns:**

["NO or VERY MINOR ethics concerns only"]

**Final Justification:**

I would like to keep my score for this work. Although the experimental results are still limited, I feel that the paper has a novel theoretical contribution which would be useful for the community

**Limitations:**

Yes

**Quality:**

3

**Strengths And Weaknesses:**

Strengths
- I find it interesting that this local search operations on an already learned tree is such a common practice but no one has studied this from a fixed parameter tractability perspective. Hence, I feel this work bridges this gap.
- The paper is overall well written but the diagrams like figure 1, 2 are confusing and difficult to understand.

Weaknesses
-  This work does not explore some dataset settings where their theoretical results could be more practical like when are each of the parameters for which the algorithms are tractable smaller and hence in practice, the results hold.
- The authors have not discussed the technical hardness/novelty of their hardness results and what are the key challenges in obtaining the results.

---

> ### Author Rebuttal · Authors · 2025-07-31
>
> Thank you for the review and the feedback! We respond to the Reviewer's concerns and questions separately below.
>
> **Figures 1 and 2**
>
> - We will try to make the figures clearer by adding explanations to the paper along the lines of what follows:
>
> - In Figure 1, the parameterized tractability of the problems increases as one traverses the Hasse diagram upwards. For example, both problems are W[1]-hard and in XP when parameterized by $s$, and consequently they are at least W[1]-hard (potentially paraNP-hard) for all parameters smaller than $s$ and have at least an XP-time algorithm (potentially FPT) for all parameters larger than $s$. The leftmost cell of each tells the complexity classes for threshold adjustment and the rightmost for cut exchange, which happen to be the same for all considered single parameters.
>
> - Figure 2 is admittedly very information-dense as it concisely summarizes all our results for parameter combinations. There, the result implied by the coloring of the box applies to all parameters bounded from below by a function of the parameter in the bottom cell and from above by a function of the parameter in the top cell. As a concrete example, the first box of the middle row states that Threshold Adjustment is W[1]-hard and in XP for, for example, $k$, $k + D$, $k + t$, and $k + D + t$.
>
>
>
> **Used data sets**
>
> - Our data sets are standard benchmarks used in many other papers on learning/optimizing decision trees (Bessiere et al., 2009; Narodytska et al., 2018; Romano et al., 2022; Staus et al., 2025).
>
> - The most practical algorithm and the one used in our experiments seems to be the one whose running time depends especially on the number of features $d$ and the maximum domain size $D$, which both range from a few to hundreds in the test instances and thus cover a variety of settings.
>
> **Technical hardness**
>
> - While some of our results are adaptations from previous work, we wish to emphasize that many others are relatively involved; consider, e.g., the proofs of Theorem 3.6 (Threshold Adjustment reduces to Cut Exchange) or Theorem 5.2 (W[1]-hardness for $d$), requiring meticulous analysis and designing novel gadgets. We will update the manuscript accordingly to highlight our technical contributions more.
>
> **Are features of descendants modified?**
>
> - No, both operations only apply to the chosen cut.
>
> **Other operations**
>
> - The parameterized complexity of removing leaves is well-understood (Harviainen et al., 2025), and we also include those operations in some of our results: see Theorem 4.3 and the experiments, in particular Appendix E. One can also simulate pruning operations by threshold adjustment (and thus also cut exchanges), since setting the threshold to an arbitrarily large/small value essentially removes one of the subtrees. To keep the scope of the paper focused, we did not consider other operations here, but those would provide an interesting future research direction.
>
> **Near-optimality locally**
>
> - On lines 67–72, we discuss that the surprisingly good performance of heuristics for learning and pruning. In particular, the constructed trees are near-optimal in the local neighborhood, which in this case is defined as the set of decision trees obtainable by performing $k$ threshold adjustments/cut exchange operations. On some instances, there are better trees in the local neighborhood of the tree obtained from heuristics, but the improvements are mild. For example, with the *breast-cancer* instance, performing one adjustment/exchange reduces the number of errors from 31 to 30. With 'mild' we mean that the error reduction is quite small.
>
> **Popularity of local search**
>
> - We suspect that the contrast between the limited improvement observed in our experiments and the popularity of local search is at least partially explained by the quality of the decision tree that is used initially. When local search or genetic algorithms are applied in practice, perhaps the initial decision tree tends to be mediocre and thus easy to improve at first, whereas better heuristics produce a near-optimal tree that is harder to improve.
>
> **Improving the running time**
>
> - Our proof-of-concept implementation is written in Python and loops over all the dynamic programming table entries, which can be inefficient in Python. In addition to implementing the algorithm in a faster language like C++, one could look for heuristic improvements like avoiding enumerating unnecessary table entries.
>
> - Even though the exponential time hypothesis (ETH) incurs some restrictions on the optimal asymptotical time complexities, there is also room for improving the exponents in the running times. More precisely, currently the exponent is $2d$, but the ETH only excludes an exponent of $o(d)$.

---

### Decision · Program_Chairs · 2025-09-17

**Decision:**

Accept (poster)

**Comment:**

This paper studies the problem of improving decision trees through a fixed number of local-search operations, specifically threshold adjustment at internal nodes and feature exchange with new thresholds. The authors show that both problems are NP-complete in general, but provide a detailed parameterized complexity analysis that identifies tractable cases, such as when both the number of features and feature domain size are small, yielding fixed-parameter tractability (FPT). They extend existing hardness results, formally connect the two problems, and present FPT and XP algorithms under various parameter combinations. Empirical results on benchmark datasets indicate that standard greedy decision tree learners are already near-optimal in their local neighborhoods, with local modifications yielding only marginal improvements, offering insight into the challenges faced by local search–based decision tree construction methods.

In general the main positive aspects about the paper is that it addresses an interesting question and also makes theoretical contributions.

On the negative side, we find that the usefulness is rather limited. Partially, a significant computational effort need to be made to obtain marginal improvements. The utility/cost ratio seems to be a bit poor.

Looking at the discussion, the group of reviewers is a bit divided, but I find that the arguments of the reviewers who recommend rejection are a bit subjective and not sufficiently strong to prevent the work from being presented. Therefore, I will recommend acceptance as a poster.

This being said, I find the authors' statement that "marginal improvement is a strength of this paper" quite disturbing. If this is what you are after, this should maybe be the main message and focus of the paper.